# Ubiquitin-specific protease 21 stabilizes BRCA2 to control DNA repair and tumor growth

Jinping Liu[1,2], Alex Kruswick[1], Hien Dang[2], Andy D. Tran[1], So Mee Kwon[2], Xin Wei Wang[2] & Philipp Oberdoerffer[1]

Tumor growth relies on efficient DNA repair to mitigate the detrimental impact of DNA damage associated with excessive cell division. Modulating repair factor function, thus, provides a promising strategy to manipulate malignant growth. Here, we identify the ubiquitin-specific protease USP21 as a positive regulator of BRCA2, a key mediator of DNA repair by homologous recombination. USP21 interacts with, deubiquitinates and stabilizes BRCA2 to promote efficient RAD51 loading at DNA double-strand breaks. As a result, depletion of USP21 decreases homologous recombination efficiency, causes an increase in DNA damage load and impairs tumor cell survival. Importantly, BRCA2 overexpression partially restores the USP21-associated survival defect. Moreover, we show that USP21 is overexpressed in hepatocellular carcinoma, where it promotes BRCA2 stability and inversely correlates with patient survival. Together, our findings identify deubiquitination as a means to regulate BRCA2 function and point to USP21 as a potential therapeutic target in BRCA2-proficient tumors.

---

[1] Laboratory of Receptor Biology and Gene Expression, National Cancer Institute, 41 Library Drive, Bethesda, MD 20892, USA. [2] Laboratory of Human Carcinogenesis, National Cancer Institute, Building 37, Bethesda, MD 20892, USA. Correspondence and requests for materials should be addressed to P.O. (email: Philipp.Oberdoerffer@nih.gov)

Accurate genome maintenance ensures cell integrity and survival by minimizing genetic and epigenetic defects. Dividing cells are particularly at risk, as DNA replication is a major source of DNA damage, which can result in cell cycle arrest, aberrant mitosis and cell death if not properly repaired. Defects in DNA repair are further linked to genomic aberrations that can promote malignant transformation[1, 2]. Paradoxically, DNA repair is also essential for tumor cell survival, and cancer cells invariably adjust their DNA damage response (DDR) to deal with the DNA damage load associated with excessive cell division[3–5]. The identification of factors that modulate DNA repair efficiency is, thus, emerging as a viable strategy to manipulate (cancer) genome maintenance and, thereby, tumor cell survival.

A central aspect of genome integrity in dividing cells is DNA repair via homologous recombination (HR). HR is a conserved and generally error-free mechanism to eliminate DNA double-strand breaks (DSBs) and is essential for the resolution of arrested DNA replication forks, thus ensuring successful S phase progression. HR is generally initiated by the PI3-like kinases ATM and ATR, which are activated by DSBs and stalled replication forks, respectively, to induce a cascade of post-translational phosphorylation events, including the formation of S139-phosphorylated histone H2AX (γ-H2AX) at sites of DNA damage. The latter facilitate the assembly of downstream HR effectors, most notably the breast and ovarian tumor suppressors BRCA1, BRCA2 and PALB2, which, together, promote DSB end resection and the formation of RAD51-coated single-stranded DNA (ssDNA) filaments required for homology search (reviewed in ref. [2]). Consistent with their essential role in HR, deleterious mutations in BRCA proteins or PALB2 promote genome maintenance defects that lead to chromosomal aberrations and, consequently, malignant transformation[6]. On the other hand, increased expression of either RAD51 or BRCA2 have been observed in several tumor types and were proposed to accommodate for repair requirements associated with DNA replication[7, 8]. In support of the latter, ovarian tumors with intact BRCA genes were found to be associated with a significantly higher likelihood of poor survival than tumors with BRCA mutations[9]. Conversely, BRCA-deficient tumors are uniquely sensitive to replication stress-inducing genotoxic drugs[10, 11]. Together, these findings emphasize the central role for HR proteins during both normal and malignant cell division.

The function and stability of DSB repair factors is tightly regulated by post-translational modifications. In recent years, ubiquitination—the covalent attachment of a 76 aa ubiquitin (Ub) protein to target molecules—has emerged as a central DDR modulator[12]. Ubiquitination encompasses a sequential enzymatic reaction mediated by E1, E2 and E3 ligases, which results in mono- or poly-ubiquitinated lysine residues on-target proteins. Lys 48-linked poly-Ub chains target substrates to proteasome-dependent degradation, whereas other types of (poly-)ubiquitination can play roles in the control of protein interactions, activity, subcellular localization and scaffolding[13]. Ubiquitination is often regulated by its removal through the actions of specific deubiquitinating enzymes (DUBs), of which ubiquitin-specific proteases (USPs) comprise the largest sub-family (~60 genes)[5]. Many E2/E3 ligases and DUBs have now been linked to DSB repair[5, 14–16]. Of relevance for HR, (de)ubiquitination events were found to directly or indirectly modulate the function or stability of RAD51, CtIP, BRCA1, BRCA2 and PALB2[17–24]. BRCA2 protein levels were further reported to correlate inversely with Skp2 E3 ligase expression in prostate tumor tissue[25], and BRCA2 stabilization has been linked to sporadic breast cancer development[7]. However, both the mechanistic basis and physiological relevance of these observations remain to be investigated.

Taken together, ubiquitination is emerging as a central rheostat for HR capacity, which may have direct implications for malignant transformation and/or tumor growth[5], particularly in the absence of apparent genetic defects in DDR components. Here, we identify the DUB enzyme USP21 as an HR-associated modulator of tumor cell survival. USP21 facilitates HR at least in part by stabilizing BRCA2 protein levels and, concomitantly, promoting RAD51 recruitment to DSBs. Importantly, we find that USP21 is the most highly amplified DUB in hepatocellular carcinoma (HCC), a BRCA2-proficient tumor with poorly understood molecular pathways of carcinogenesis, a scarcity of druggable targets and a dismal therapeutic outcome[26]. USP21 stabilizes BRCA2 in patient-derived HCC tumor cell lines, protects from DNA damage and promotes tumor cell growth in a BRCA2-dependent manner. Consistent with this, high USP21 expression levels correlate with poor HCC patient survival. USP21 thus presents a mediator of genome maintenance that may serve both as a therapeutic target and potential biomarker for HCC.

## Results

**USP21 promotes DSB repair via HR.** Using an RNA interference-based screen for HR modulators, we have previously identified USP21 as a potential positive effector of HR[27] (Supplementary Fig. 1A). To validate the primary screen results and further dissect the impact of USP21 on DSB repair, we performed both overexpression and knockdown of USP21 in a DRGFP-based HR reporter cell line, which allows for doxycycline (Dox)-dependent induction of I-SceI endonuclease-mediated DSBs[27]. Consistent with an HR-promoting role, USP21 overexpression increased HR efficiency, and USP21 depletion with two independent shRNAs resulted in a pronounced decrease in HR compared to a luciferase-specific control shRNA (sh-Luc) (Fig. 1a, Supplementary Fig. 1B). While USP21 loss caused a modest reduction in S/G2 phase cells (Supplementary Fig. 1C), the latter was not sufficient to account for the observed HR defect, suggesting a direct role for USP21 in homologous repair. To independently validate our knockdown results, we generated USP21 knockout U2OS HR reporter cells using CRISPR/Cas9-mediated gene editing. USP21 deletion was confirmed both at the genomic and protein levels, and HR efficiency was significantly reduced in two independent USP21 KO clones, with little to no changes in cell cycle profiles (Fig. 1b, Supplementary Fig. 2A–E). Of note, USP21-deficient clones recovered from the HR defect after extended time in culture despite stable USP21 deletion (Supplementary Fig. 2C, D), pointing to compensatory mechanisms that may counteract deleterious consequences of USP21 loss, as reported previously in mice[28, 29]. Given that compensatory networks were found to be activated in response to deleterious mutations rather than gene knockdown[30], subsequent experiments were performed under conditions of acute USP21 depletion via RNAi, or in early-stage USP21 KO cells.

Since ubiquitin modifications as well as several DUBs and E3 ligases have been implicated in both HR and non-homologous end joining (NHEJ)[5], we next asked if the effect of USP21 loss reflected a general DSB repair defect. Using a GFP-based NHEJ reporter[31], we found that depletion of USP21 with either siRNA or shRNA had only a minor effect on NHEJ, corroborating a pathway-specific role for USP21 in DSB repair (Fig. 1c, Supplementary Fig. 1D, E). Depletion of 53BP1, a factor implicated in end joining processes, served as a control[2]. Consistent with the HR defect observed upon USP21 depletion, we further detected increased γ-H2AX accumulation following laser-induced DSB formation in S phase cells (Fig. 1d). Together, these findings demonstrate a role for USP21 in HR.

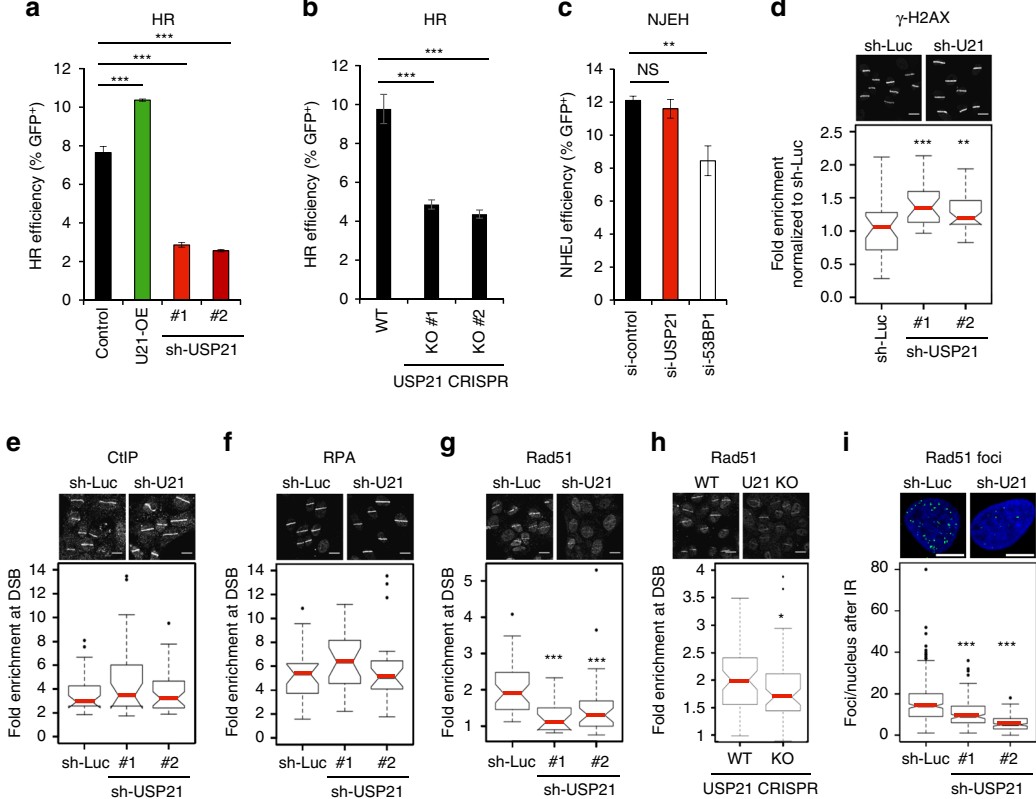

**Fig. 1** USP21 promotes HR by facilitating RAD51 accumulation at DSBs. **a** HR efficiency in TRI-DRGFP reporter cells in the presence of the indicated shRNAs or transient USP21 overexpression. One of at least three representative experiments is shown. **b** HR efficiency in control (WT) TRI-DRGFP cells and two independent CRISPR/Cas9 USP21 knockout (KO) clones. **c** NHEJ efficiency in pEJ5-GFP-transgenic U2OS cells transfected with the indicated siRNAs. Values in **a** through **c** are expressed as mean and s.d. and samples were analyzed in triplicate. **d–h** Immunofluorescence (IF) analyses at laser-induced DSBs in U2OS cells 1 h after release from double-thymidine-block. Representative images are shown, *scale bar*: 20 μm. For *box plots*, bottom and top hinges denote first and third quartile, respectively. *Whiskers* denote ±1.5 interquartile range (IQR). The *red horizontal line* represents the median. *p*-values are based on a two-sided Wilcoxon rank-sum test, sh-U21: sh-USP21. **d** γ-H2AX levels in cells expressing sh-Luc ($n = 30$), sh-U21-1 ($n = 26$) or sh-U21-2 ($n = 35$). One of five representative experiments is shown. Fluorescence intensity of γ-H2AX$^+$ areas was quantified and normalized to sh-Luc. **e** CtIP accumulation in cells expressing sh-Luc ($n = 28$), sh-U21-1 ($n = 27$) or sh-U21-2 ($n = 36$). Fold enrichment reflects the ratio of γ-H2AX$^+$ over γ-H2AX$^−$ nuclear areas. **f** RPA accumulation at laser-induced DSBs as in **e**; sh-Luc ($n = 23$), sh-U21-1 ($n = 22$), sh-U21-2 ($n = 23$). **g** RAD51 accumulation at laser-induced DSBs as in **e**. One of five representative experiments is shown, sh-Luc ($n = 30$), sh-U21-1 ($n = 32$), sh-U21-2 ($n = 37$). **h** RAD51 accumulation as in **g** using control (WT, $n = 60$) and USP21 KO cells ($n = 55$). **i** IF analysis of RAD51 foci 6 h after IR (10 Gy). *Box plots* represent the number of nuclear RAD51 foci (*green*); sh-Luc ($n = 284$), sh-U21-1 ($n = 145$), sh-U21-2 ($n = 174$). Representative analyses and images are shown for one of two independent experiments, *scale bar*: 10 μm. Unless noted otherwise, *p*-values are based on Student's two-tailed *t*-test: *$p < 0.05$, **$p < 0.01$, ***$p < 0.001$

**USP21 affects RAD51 loading but not end resection at DSBs.** To gain mechanistic insight into USP21 function during HR, we performed a systematic dissection of HR-relevant DSB repair events starting with DSB-proximal ubiquitination. The latter is required for the recruitment of a functionally diverse set of DSB repair factors, including mediators of HR (i.e., BRCA1) and NHEJ (i.e., 53BP1)[32–34]. An analysis of DSB-induced poly-Ub chain formation at sites of irradiation-induced DNA damage revealed a modest and transient increase upon USP21 depletion (Supplementary Fig. 3A). Consistent with the latter, USP21 loss resulted in an equally modest increase in the accumulation of 53BP1 and BRCA1 (Supplementary Fig. 3B, C). USP21 contains both nuclear localization and export signals, and we could readily detect transient recruitment of a USP21-GFP fusion protein to laser-induced DSBs following nuclear enrichment with leptomycin B (LMB) (Supplementary Fig. 3D–F). These findings suggest that USP21 may act at least in part by counteracting the effects of DSB-associated E3 ligases to modulate ubiquitin-dependent repair factor accumulation at DSBs[35]. However, given that the early HR mediator BRCA1 was efficiently recruited to DSBs in the absence of USP21, these observations failed to explain the

observed HR defect. We, thus, sought to identify changes in HR events downstream of BRCA1. A central next step in the HR process is the CtIP-dependent resection of DSB ends to generate ssDNA, which is stabilized by its association with replication protein A (RPA) (reviewed in ref. [2]). End resection appeared normal in USP21-depleted cells, as both CtIP and RPA were efficiently recruited to laser-induced DSBs in S phase cells (Fig. 1e, f, Supplementary Fig. 1F). RPA is subsequently exchanged for the RAD51 ATPase to form RAD51-ssDNA filaments, which initiate homology search and strand invasion to complete HR. Notably, RAD51 recruitment to laser-induced DSBs was significantly impaired upon USP21 depletion by RNAi and in HR-defective *USP21* KO cells (Fig. 1g, h). Consistent with this, we observed fewer irradiation-induced RAD51 foci in USP21-deficient cells (Fig. 1i). Together, these findings demonstrate that USP21 facilitates HR by promoting RAD51 accumulation at DSBs.

**USP21 loss causes a decrease in BRCA2 protein levels.** As end resection was unaffected by USP21 loss, we asked if impaired

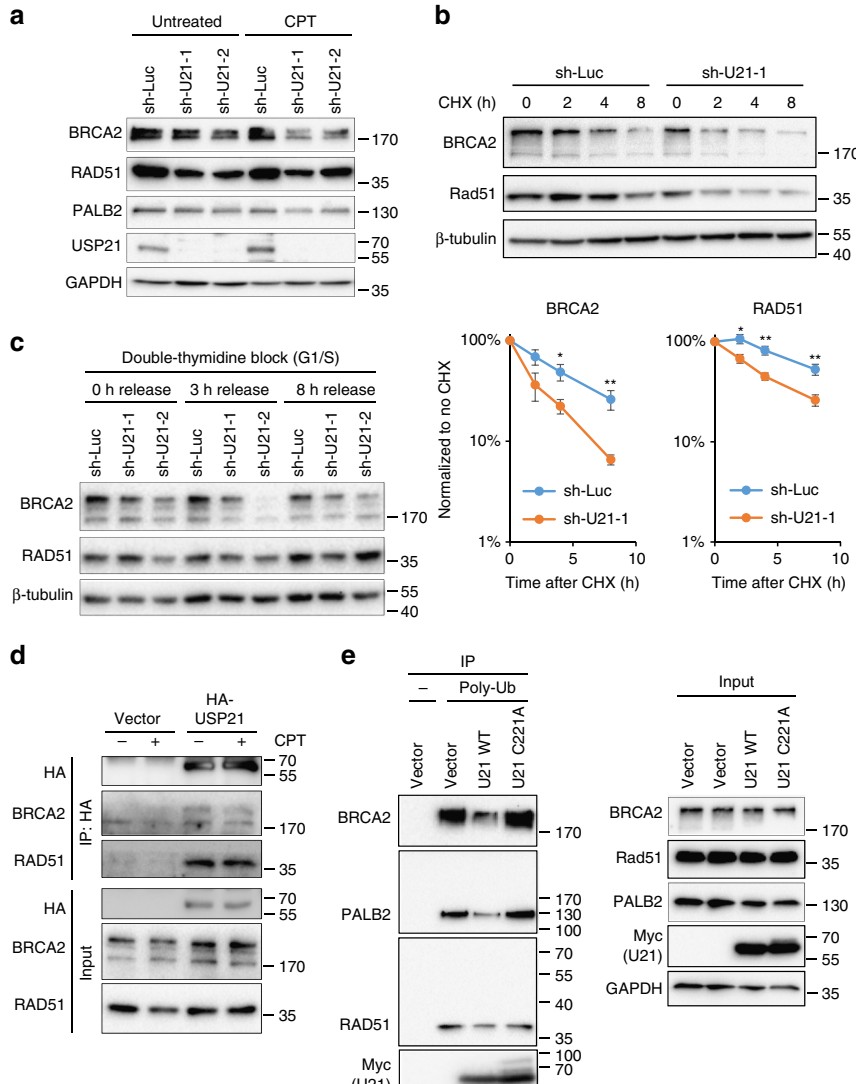

**Fig. 2** USP21 controls BRCA2 protein levels and ubiquitination. **a** Western blot analysis of the indicated proteins in the presence or absence of *USP21* knockdown. Asynchronous U2OS cells were either left untreated or treated with CPT for 1 h. One of three representative experiments is shown. **b** BRCA2 and RAD51 protein turnover in HEK293T cells in the presence of *USP21* knockdown or sh-Luc control. Western blot analysis was performed at the indicated timepoints after cycloheximide (CHX) treatment. A quantification of BRCA2 and RAD51 protein levels normalized to β-tubulin and 0 h CHX is shown. Values are expressed as mean and s.e.m. from three (2 h) or six independent experiments (4 and 8 h timepoints), *p*-values are based on Student's two-tailed *t*-test: \**p* < 0.05, \*\**p* < 0.01. **c** Western blot analysis of BRCA2 and RAD51 levels in S phase cells. Following double-thymidine-block, cells were released for the indicated time frames prior to analysis. **d** IP using anti-HA conjugated beads in HEK293T cells transiently expressing empty vector or HA-USP21 in the presence or absence of CPT. Representative immunoblots from one of three independent experiments are shown. **e** Poly-ubiquitination analyses of the indicated proteins in cells expressing empty vector, Myc-USP21 WT (U21 WT) or Myc-USP21 C221A (U21 C221A). Endogenous poly-ubiquitin-chains were immunoprecipitated in the presence of MG-132 using TUBE1. Representative immunoblots from one of two independent experiments are shown

RAD51 recruitment is due to a defect in RAD51 loading onto ssDNA, a process that critically depends on the BRCA1-interacting proteins PALB2 and BRCA2[2]. Given that BRCA2 was previously reported to be the subject of ubiquitin-mediated proteolysis[23], we sought to analyze the impact of USP21 loss on BRCA2 stability. *USP21* knockdown with two independent shRNAs caused a reduction in BRCA2 protein levels that was particularly pronounced in the presence of the topoisomerase I inhibitor camptothecin (CPT), which causes DSB induction in HR-permissive S phase cells[36] (Fig. 2a). A similar decrease in BRCA2 was observed upon treatment with the replication poison hydroxyurea (HU). Consistent with the USP21 loss-associated repair defect, decreased BRCA2 levels translated into reduced, chromatin-associated BRCA2 (Supplementary Fig. 4A). The

reduction in BRCA2 protein was not a result of transcriptional deregulation, as it was also observed in the absence of protein synthesis (Fig. 2b). Moreover, BRCA2 mRNA levels remained unaltered upon USP21 depletion (Supplementary Fig. 4B). Together, these findings point to USP21 as a mediator of BRCA2 stability. We further detected a decrease in RAD51 protein, which is in agreement with a previously reported role for BRCA2 in stabilizing RAD51 and was also observed upon siRNA-mediated depletion of BRCA2 (Fig. 2a, b, Supplementary Fig. 4C)[24, 37]. Of note, the reduction in RAD51 protein by itself did not account for impaired RAD51 accumulation at DSBs, as DSB-associated RAD51 levels were normalized to overall nuclear RAD51 abundance and, thus, reflect a bona-fide RAD51 recruitment defect (Fig. 1g, h). PALB2 protein levels remained

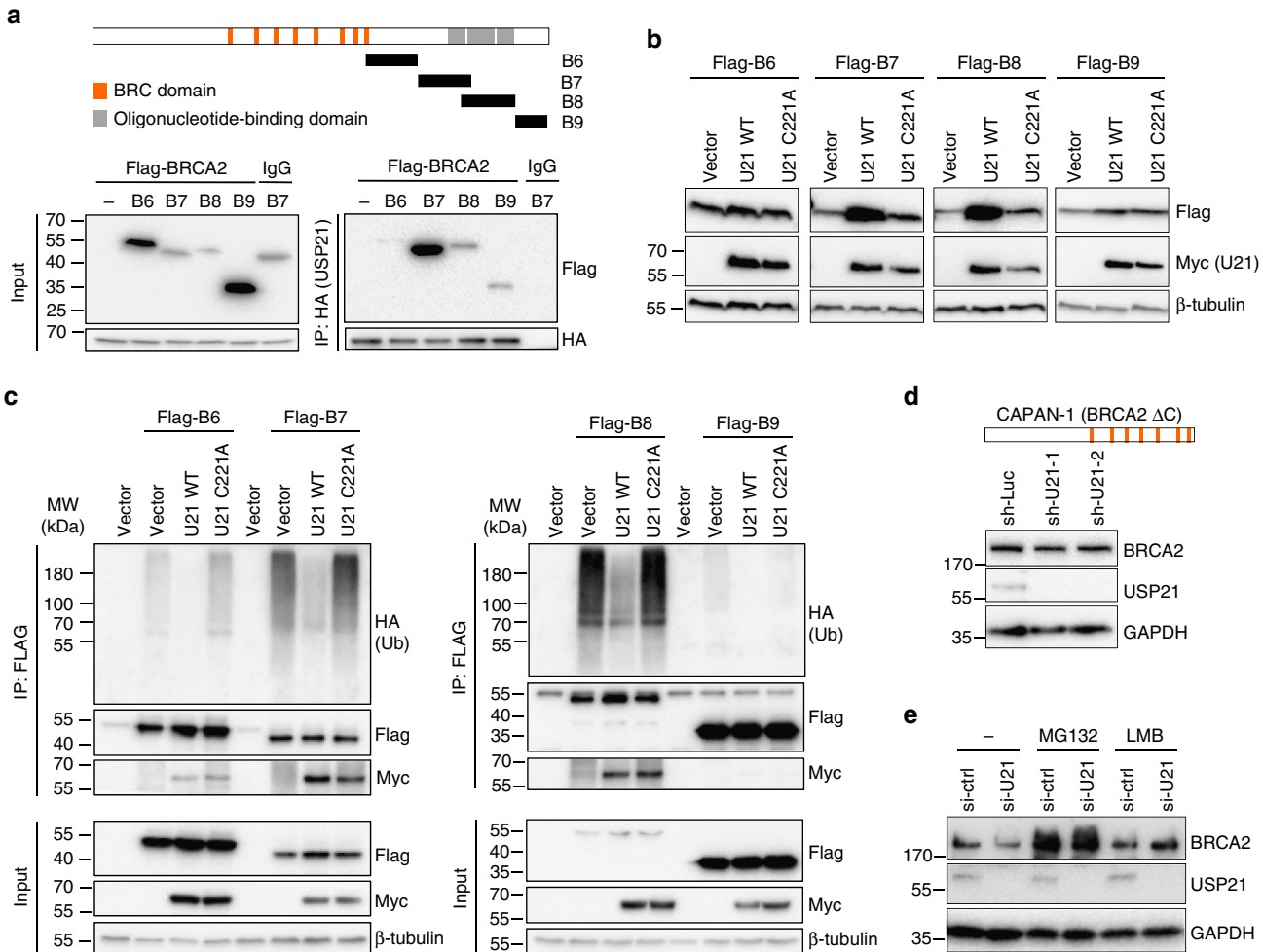

**Fig. 3** USP21 deubiquitinates and stabilizes BRCA2 via its C-terminus. **a** IP using anti-HA conjugated beads in HEK293T cells expressing HA-USP21 and the indicated Flag-BRCA2 fragments. A schematic of full-length BRCA2 and BRCA2 C-terminal fragments is shown. **b** Western blot analysis of the indicated BRCA2 fragments in the presence of empty vector, Myc-USP21 WT (U21 WT), or the Myc-USP21 C221A catalytic mutant (U21 C221A). **c** Ubiquitination assay in cells expressing HA-Ub and the indicated Flag-BRCA2 fragments in the presence of empty vector (V), U21 WT or U21 C221A. Cells were treated with MG-132 prior to Flag-IP. **d** BRCA2 western blot analysis in CAPAN-1 cells in the presence or absence of *USP21* knockdown. A schematic for the CAPAN-1-specific BRCA2 C-terminal truncation (BRCA2 ΔC) is shown. **e** BRCA2 western blot analysis of cell expressing the indicated siRNAs in the presence or absence of MG-132 or LMB treatment

largely unaltered, although a modest reduction could be observed with one of the two hairpins following CPT treatment. Given that the effect of USP21 loss on BRCA2 stability was enhanced in the presence of the S phase poisons CPT and HU, we next determined BRCA2 expression specifically in S phase cells and found that USP21 depletion resulted in a particularly robust BRCA2 decrease in this HR-prone cell cycle stage (Fig. 2c, Supplementary Fig. 1F). A correlation with RAD51 protein levels was less apparent, pointing to BRCA2 as the primary USP21 target within the BRCA2/RAD51 complex. Consistent with this, reduced BRCA2 protein levels were confirmed in USP21 KO cells (Supplementary Fig. 2F), whereas little to no change in RAD51 was detected.

**USP21 interacts with and deubiquitinates BRCA2**. We next asked whether USP21-mediated BRCA2 stabilization involves USP21 interaction with the BRCA2/RAD51 complex. Immuno-precipitation (IP) of HA-tagged USP21 revealed binding to BRCA2 as well as RAD51 at a ratio reflecting the previously reported BRCA2/RAD51 complex stoichiometry of up to six RAD51 molecules per one BRCA2 molecule (Fig. 2d)[38]. To

determine whether USP21 can promote the deubiquitination of endogenous BRCA2 and/or the BRCA2-associated RAD51 and PALB2 proteins[39, 40], we measured the extent of (poly-)ubiquitin modifications on either protein in the presence or absence of USP21 overexpression. A USP21 mutant carrying a point mutation in its reactive cysteine (USP21 C221A) served as a catalytically inactive control. Using a tandem ubiquitin-binding entity (TUBE) directed against poly-Ub moieties, BRCA2 was readily immunoprecipitated in cells treated with the proteasome inhibitor MG-132, along with PALB2 and RAD51 (Fig. 2e). However, neither PALB2 nor RAD51 showed detectable poly-ubiquitination, consistent with the notion that TUBE-based co-purification of these proteins was the result of complex formation with ubiquitinated BRCA2 (Fig. 2e). The latter is in agreement with previous work demonstrating that BRCA2 is significantly more susceptible to proteasomal degradation than its interactor PALB2[24]. Direct ubiquitination of endogenous BRCA2 was further independently confirmed using BRCA2 IP under denaturing conditions followed by detection of HA-Ub (Supplementary Fig. 5A). Importantly, overexpression of wild-type (WT) USP21 but not the catalytic mutant resulted in a marked reduction of TUBE-associated BRCA2, as well as its

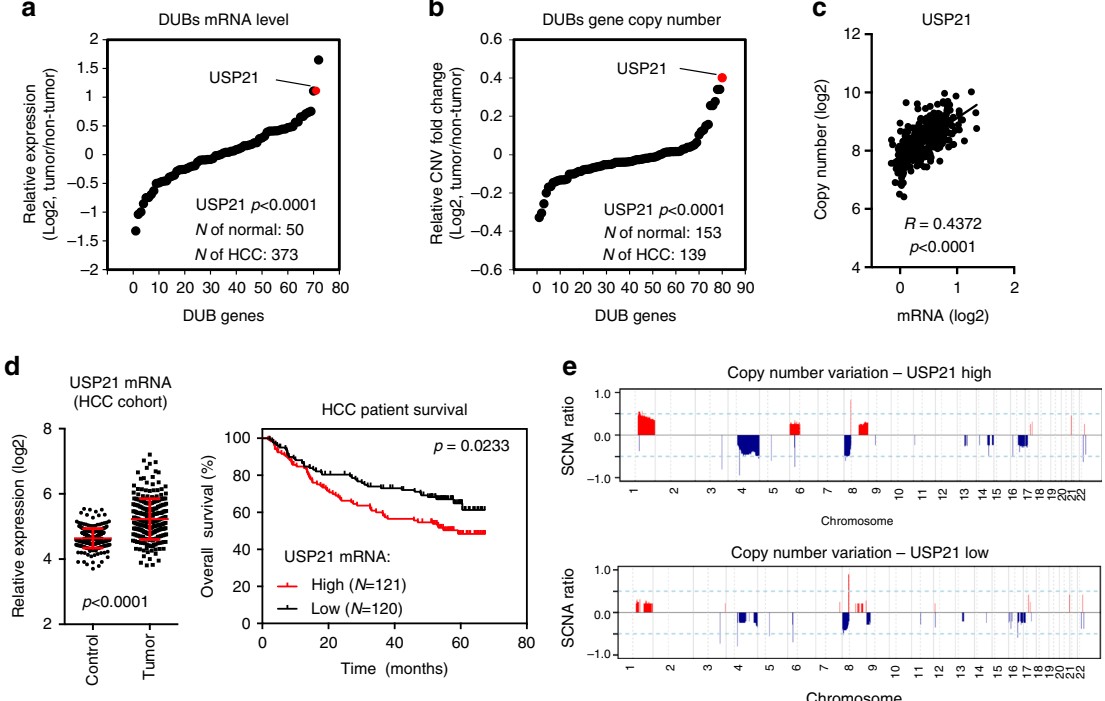

Fig. 4 USP21 is amplified and correlates with poor survival in HCC. **a** Relative expression of 72 DUBs in HCC, based on TCGA RNA-sequencing data. Expression was normalized to non-tumor tissue and is expressed as Log2 mean. USP21 is indicated in *red*. **b** Relative gene copy number change (mean, Log2) of 80 DUBs in HCC, relative to non-tumor tissue, based on TCGA data. USP21 is indicated in *red*. **c** Correlation between copy number variation and USP21 expression change in tumor compared to non-tumor tissue, based on TCGA data from **a** and **b**. **d** USP21 expression correlates with poor survival. *Left* panel: relative USP21 expression levels in the LCI cohort of 488 patient samples in HCC and non-tumor tissue, based on microarray data from[26]. *Right* panel: Kaplan-Meier survival analysis of the same patients, separated by USP21 expression. USP21 low reflects the bottom 50 percent, USP21 high the top 50 percent of patients. **e** Frequencies of somatic copy number alteration (SCNAs, Log2) are plotted as a function of genome location for the clinical specimen from **d**, separated in USP21 high or low groups. Chromosome boundaries and centromere position are indicated by vertical *solid* and *dashed lines*, respectively. Horizontal *dashed blue lines* indicate frequencies of >±0.5 (Log2). Unless noted otherwise, *p* values are based on Student's two-tailed *t*-test

interactors PALB2 and RAD51 (Fig. 2e). Taken together, these findings point to BRCA2 as a target for USP21-mediated deubiquitination in vivo.

We next sought to determine whether BRCA2 is a bona-fide catalytic substrate for USP21. BRCA2 protein stability was found to be regulated via its C-terminal oligonucleotide-binding (OB) domains[41, 42]. Consistent with this, co-IP of HA-tagged USP21 with a set of Flag-tagged C-terminal BRCA2 fragments[43] revealed preferential interaction with OB domain-containing fragments (B7 and B8) when compared to two OB domain-flanking fragments (B6 and B9, Fig. 3a). To determine whether USP21 interaction can promote the stabilization of BRCA2 fragments, we expressed the latter in the presence of either WT USP21 protein or the catalytically dead C221A mutant. Notably, USP21 overexpression caused a significant, DUB activity-dependent increase in BRCA2 fragment levels that correlated with the extent of USP21 interaction and was most pronounced for the OB domain-containing constructs (Fig. 3b). We next asked if USP21-mediated BRCA2 fragment stabilization is associated with changes in poly-ubiquitination. Consistent with this, IP of BRCA2 fragments in the presence of HA-tagged Ub revealed robust poly-Ub modification, which was most pronounced on those fragments found to strongly interact with USP21 (B7 and B8), and was significantly reduced following overexpression of WT, but not the catalytic dead USP21 protein (Fig. 3c). To determine whether USP21 is sufficient to deubiquitinate BRCA2 fragments, we performed in vitro deubiquitination assays using HA-ubiquitinated, immuno-purified BRCA2 fragments as well as

full-length Flag-BRCA2. Incubation with recombinant USP21 protein resulted in complete removal of HA-Ub (Supplementary Fig. 5B, C). Purified fragments did not contain DUB activity as incubation in the absence of USP21 did not affect HA-Ub levels (Supplementary Fig. 5D). Together, these findings demonstrate that USP21 can interact with, deubiquitinate and stabilize BRCA2, at least in part via its C-terminal OB domains. Further corroborating the involvement of the BRCA2 C-terminus, USP21 depletion did not alter BRCA2 protein levels in CAPAN-1 pancreatic tumor cells, which express a C terminally truncated, hypomorphic BRCA2 protein lacking the OB domains (Fig. 3d).

Our data collectively point to a mechanism of USP21-mediated BRCA2 stabilization via proteasomal degradation. Consistent with this, inhibition of proteasome activity was able to stabilize BRCA2 and reverse the reduction in BRCA2 protein levels following USP21 depletion (Fig. 3e). Given that USP21 is largely cytoplasmic (Supplementary Fig. 3E), and in light of the finding that BRCA2 function can be regulated by nucleo-cytoplasmic shuttling[44], we asked if nuclear export may contribute to BRCA2 destabilization. Consistent with this notion, we found that inhibition of nuclear export was able to reverse the reduction in BRCA2 protein levels in USP21-depleted cells (Fig. 3e). Altogether, these findings identify USP21 as a regulator of BRCA2 protein turnover.

**USP21 is a prognostic marker of HCC.** Having established a role for USP21 in stabilizing BRCA2, we next sought to investigate the

physiological relevance of this finding. The stabilization of DDR factors involved in HR and other replication stress-related repair processes was proposed to play an important role in promoting the survival of rapidly dividing tumor cells[8, 45, 46]. Thus, we sought to determine whether *USP21* expression is elevated in tumors. A TCGA survey of *USP21* mRNA expression revealed that 7 of the 26 interrogated cancer types showed significantly elevated *USP21* levels when compared to non-tumor tissues ($p < 0.001$, Student's two-tailed $t$-test) (Supplementary Fig. 6A). The most pronounced increase in *USP21* mRNA was observed in liver cancer, including cholangiocarcinoma and hepatocellular

carcinoma (HCC). With the exception of p53, HCC carries no apparent genetic defects in DDR signaling[47, 48]. The stabilization of HR factors may, thus, present a plausible strategy to enhance repair efficiency in these tumors.

To determine whether increased DUB expression is a general feature of HCC, we analyzed the expression of all expressed DUBs in the TCGA HCC patient data set. Only three DUBs, including *USP21*, showed a greater than twofold increase in expression compared to non-tumor tissue (Fig. 4a). Moreover, copy number variation (CNV) analyses revealed *USP21* as the most highly amplified DUB in HCC ($p = 2.86131 \times 10^{-40}$, Student's two-tailed

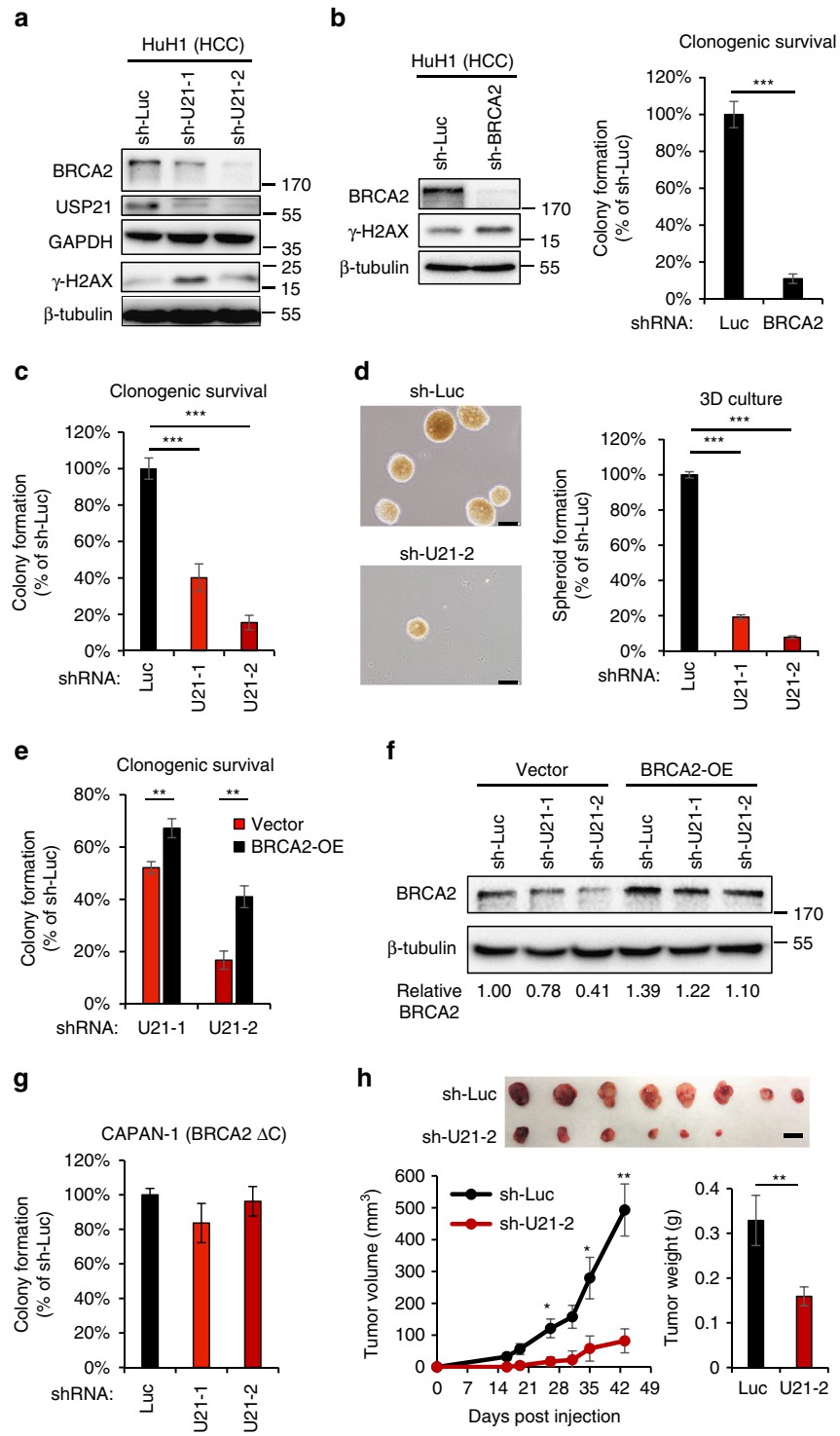

*t*-test, Fig. 4b). *USP21* CN was significantly correlated with *USP21* mRNA expression ($r = 0.4372$, Fig. 4c), suggesting that increased *USP21* expression may be the result of aberrant *USP21* gene amplification. Increased *USP21* expression was validated in an independent cohort of 488 patients (Fig. 4d)[26]. Importantly, elevated *USP21* expression was correlated with poor survival in this cohort (Fig. 4d), and a similar relationship was observed in an independent patient group (Supplementary Fig. 6B). Moreover, increased *USP21* mRNA expression was correlated with somatic copy number alterations (SCNAs) in patients with high compared to low *USP21* expression (Fig. 4e). Notably, tumor-associated SCNAs often result from non-allelic homologous recombination events[49], consistent with our observation that USP21 positively regulates HR. Together, these findings point to a unique, tumor-promoting role for USP21 in HCC.

**USP21 controls BRCA2 levels and tumor cell survival in HCC.** To gain mechanistic insight into USP21 function in HCC, we asked whether USP21 can modulate BRCA2 stability in HCC-derived tumor cells[50]. Consistent with our previous findings, *USP21* knockdown in hepatoma cell lines resulted in reduced BRCA2 expression and a concomitant increase in DNA damage, as measured by H2AX phosphorylation (Fig. 5a, Supplementary Fig. 7A). BRCA2 was previously shown to be essential for the survival of highly proliferative cell types, such as embryonic stem cells and embryonic tissue, which has been attributed to its impact on genome maintenance in S phase[51]. Extending these observations to tumor cell growth, we found that shRNA-mediated *BRCA2* knockdown causes a significant reduction in the clonogenic survival of HuH1 hepatoma cells (Fig. 5b). We thus reasoned that USP21 depletion and the associated reduction in BRCA2 may result in a similar tumor growth defect. Consistent with this, we observed a pronounced decrease in the clonogenic survival of HuH1 cells following stable *USP21* knockdown, which further correlated with the extent of BRCA2 reduction (Fig. 5c). Similar results were obtained with another HCC cell line (Hep3B, Supplementary Fig. 7B). USP21 loss caused a variable induction of caspase 3 cleavage and/or a moderate reduction in S phase cells, suggesting that a combination of apoptotic cell death and cell cycle arrest contributes to the observed growth defect (Supplementary Fig. 7C–F). To better recapitulate physiological tumor growth conditions, we determined HCC oncosphere formation using an AlgiMatrix-based 3D organoid culture model previously reported to effectively capture HCC aggressiveness in vitro[52]. Consistent with the hypothesis that USP21 is oncogenic, sphere formation was significantly inhibited upon USP21 depletion (Fig. 5d).

To determine if the reduction in the growth of USP21-depleted tumor cells is functionally linked to the decrease in BRCA2 protein, we generated stable, *BRCA2*-transgenic HCC cell lines and followed clonogenic survival in the presence or absence of *USP21* knockdown. BRCA2 overexpression was able to partially restore colony formation in USP21-depleted hepatoma cells, supporting a role for BRCA2 as a mediator of USP21-dependent tumor growth (Fig. 5e, f, Supplementary Fig. 7G). Consistent with the latter, USP21 loss did not significantly alter the survival of CAPAN-1 prostate cancer cells, which express a hypomorphic BRCA2 protein lacking its USP21-interacting C-terminal domain (Figs. 3d and 5g). Together, these findings support the notion that USP21 promotes BRCA2 stability and protects from DNA damage accumulation in BRCA2-proficient tumor cells, which can in turn contribute to increased tumor growth and, ultimately, a more malignant phenotype. Consistent with the latter, USP21 loss impaired HuH1 tumor growth in vivo in a xenograft assay (Fig. 5h).

## Discussion
Here, we identify the ubiquitin-specific protease USP21 as mediator of HR. USP21 stabilizes BRCA2 and promotes RAD51 recruitment to DSBs. Importantly, we provide evidence that a defect in USP21-mediated BRCA2 stabilization impairs the growth of BRCA2-proficient HCC tumor cells, and consistent with this, *USP21* levels inversely correlate with HCC patient survival, pointing to USP21 as a potential target for HCC tumor therapy.

BRCA2 is an essential mediator of genome maintenance in S phase cells, which has direct implications for replication stress - a major impediment to tumor growth[3, 4]. However, little is known about the factors that regulate BRCA2 stability. We now show that USP21 interacts with and deubiquitinates BRCA2 and that USP21 loss results in decreased BRCA2 expression in tumor cell lines. Biochemical dissection suggests that USP21 can associate with the C-terminal OB domains of BRCA2, and consistent with this, a tumor cell line with a C-terminal BRCA2 truncation is unresponsive to USP21 depletion with regard to BRCA2 stability and tumor growth (Figs. 3d and 5g). Notably, the OB domain was found to interact with the 70 aa small, highly acidic DSS1 protein, which is essential for HR and replication fork repair[53, 54]. Disruption of the DSS1 interacting domain or depletion of DSS1 itself results in unmasking of an NES on BRCA2, causing preferential cytoplasmic localization as well as BRCA2 protein degradation[41, 44]. Consistent with a link between the two processes, we found USP21 to be predominantly cytoplasmic, and BRCA2 destabilization upon USP21 loss could be reversed by inhibiting nuclear export. Given that the effect of USP21 on BRCA2 stability appeared most pronounced in S phase cells, it will be interesting to determine if USP21 function and/or localization are controlled in a cell cycle-dependent manner. Subcellular localization may further explain why USP21 was not identified in a recent screen for DUBs involved in DSB repair, which focused on nuclear DUBs that show robust recruitment to DSBs[15].

**Fig. 5** USP21 stabilizes BRCA2 in HCC cells to promote tumor cell growth. **a** Western blot analysis of the indicated proteins in the HuH1 hepatoma cell line in the presence or absence of *USP21* knockdown. **b** Western blot analysis and clonogenic survial of HuH1 cells in the presence or absence of *BRCA2* knockdown. Colony formation was scored in triplicate and normalized to sh-Luc. Values are expressed as mean and s.d. **c** Clonogenic survival of HuH1 cells in the presence or absence of *USP21* knockdown. Colony formation was scored in triplicate and normalized to sh-Luc. Values are expressed as mean and s.e.m. from three independent experiments. **d** HuH1 oncosphere formation in the presence or absence of *USP21* knockdown. Sphere formation was scored in triplicate and normalized to sh-Luc. Values are expressed as mean and s.d. Representative images are shown, *scale bar*: 100 μm. **e**, **f** Clonogenic survival **e** and Western blot analysis **f** after USP21 depletion in HuH1 cells stably transfected with either empty vector or Flag-BRCA2 (BRCA2-OE). The efficiency of colony formation in sh-USP21 cells was normalized to sh-Luc controls. Samples were analyzed in triplicate, values are expressed as mean and s.d. **g** Clonogenic survival of CAPAN-1 cells in the presence or absence of *USP21* knockdown. Samples are normalized to sh-Luc. Values are expressed as mean and s.e.m. from five independent experiments. A schematic of the CAPAN-1-specific BRCA2 truncation is shown. **h** HuH1 cell xenograft assay in the absence or presence of USP21. A total of 10 sites were injected per genotype and tumor size was monitored for the indicated time frame. Image and weights are from tumors isolated 6 weeks after injection, *scale bar*: 1 cm. Values are expressed as mean and s.e.m. *p*-values are based on Student's two-tailed *t*-test: *$p < 0.05$, **$p < 0.01$, ***$p < 0.001$

While BRCA2 was previously found to be ubiquitinated, USP21 is, to our knowledge, the first DUB directly involved in BRCA2 stabilization. Beyond (de)ubiquitination, BRCA2 stability was shown to depend on its interaction with PALB2, and it is tempting to speculate that USP21 may provide an additional layer of control in this context[24]. Of note, the phylogenetically related USP11 was recently reported to affect BRCA2 function in a proteasome-independent manner, by promoting the interaction of the BRCA2/PALB2 complex with BRCA1 rather than by altering BRCA2 or PALB2 protein levels[20]. Together, these observation highlight the complexity of ubiquitin-mediated HR factor regulation, and, while both protein interaction- and deubiquitination-analyses point to BRCA2 as a central target of USP21-mediated stabilization, we can at this point not rule out effects on other (poly-)ubiquitinated HR mediators. On a related note, we found that USP21 loss can lead to a decrease in RAD51 protein (Fig. 2b, Supplementary Fig. 4C). While this effect is consistent with previous reports implicating BRCA2 in RAD51 stability[37], its variable nature suggests a more nuanced relationship between the two proteins, which may at least in part be affected by compensatory mechanisms that counteract the deleterious consequences of RAD51 loss[2], and remains to be further investigated.

Underlining the physiological relevance of our findings, USP21 is significantly overexpressed in a subset of HCC patients with poor survival and correlates with increased somatic copy number alterations. Mechanistically, we show that loss of USP21 in HCC cell lines results in a reduction of BRCA2, increased DNA damage accumulation and a concomitant tumor growth defect that partially depends on BRCA2 (Fig. 5). These findings are consistent with a protective role for USP21, likely involving HR, to counteract excessive, replication-associated DNA damage, cell cycle arrest and/or apoptosis, which may, at least in part, account for the observed increase in HCC aggressiveness (poor survival). Targeting USP21 may, thus, prove to be a viable strategy to treat HCC patients, particularly in light of recent efforts directed at the development of specific small-molecule DUB inhibitors[5]. Due to the extended biological heterogeneity observed in HCC, it will, however, be important to determine to what extent USP21 and/or BRCA2 affect tumor growth across genetically distinct HCC subgroups.

The general importance of BRCA2 in the protection of cells from replication stress suggests that USP21 loss may affect additional, BRCA2-proficient tumors. Consistent with this, ~27 % of tumors showed a significant increase in USP21 expression in our TCGA analysis (Supplementary Fig. 6A). In support of a more general role in malignant transformation, overexpression of USP21 was recently found to enhance the growth of renal carcinoma cells[55], as well as teratoma formation in mice[28]. Of note, the tumor-promoting effects of USP21 are likely to go beyond the modulation of HR and/or BRCA2 stability, as BRCA2 overexpression was unable to fully restore HCC tumor cell growth (Fig. 5e). Consistent with this notion, recent work has implicated USP21 in the deubiquitination of several, potentially tumor-relevant substrates, and a role in tumor cell invasion via the modulation of cytokine expression has been reported[28, 55–57].

The Ub-mediated control of protein stability and/or function is emerging as a powerful means adapted by tumors to modulate the function of oncogenes or tumor suppressors. Inactivation of USP10 occurs frequently in tumors with wild-type p53 and was found to stabilize p53 in response to DNA damage[58]. Similarly, both the Fbx6-containing SCF E3 ubiquitin ligase and the DUB USP7 were found to modulate tumor cell survival in response to replication stress by stabilizing the ATR effector CHK1 in cancer cells and patient tissues[45, 46]. The identification of USP21 as a modulator of BRCA2 stability in HCC is, thus, expected to have important implications for both tumor growth and cancer therapy.

## Methods

**Plasmids**. Full-length *USP21* cDNA (Addgene) was PCR-amplified and subcloned into pCMV-HA or GFP-C1 vectors to generate the various constructs. The *USP21* catalytic dead C221A mutation was introduced by PCR and confirmed by DNA sequencing. Myc-USP21 constructs were a kind gift from B. Li[57]. *BRCA2* fragments B6–B9 were PCR-amplified from GFP-*BRCA2* fragments kindly provided by H. Lee[43] and subcloned into p3XFlag-CMV-10 vector. Full-length Flag-*BRCA2* was kindly provided by A. Venkitaraman[44]. Guide RNAs were inserted into the Cas9-T2A-puromycin-containing pX459 expression vector[59], see Supplementary Table 2 for guide RNA sequences.

**Cell culture and treatments**. U2OS cells (gift from M. Jasin) and HEK293T cells (gift from The Broad Institute, Cambridge, MA) were cultured in DMEM (Invitrogen) with 10% fetal bovine serum (FBS). CAPAN-1 cells (gift from S. Sharan) were cultured in IMDM with 20% FBS. Hep3B and HuH1 hepatoma cell lines were described previously[50], Hep3B cells (American Type Culture Collection) were cultured in DMEM with 10% FBS and 10% of nonessential amino acids. HuH1 cells (Japan Cell Repositories) were cultured in DMEM without pyrimidine with 10% FBS. Cell lines were negative for mycoplasma. Plasmid transfections were performed using Lipofectamine 2000 transfection reagent (Invitrogen). Lentiviral infection was carried out by spin infection (2250 rpm, 90 min, Beckman-Coulter Allegra X-12R centrifuge) with 8 µg ml⁻¹ polybrene (Sigma), cells were incubated overnight prior to virus removal and selection with puromycin (1–2 µg ml⁻¹)[27]. Individual MISSION shRNA-expressing lentiviral vectors were from Sigma (Supplementary Table 2). SiRNAs (ON-TARGET PLUS library, Dharmacon) were transfected using DF-1 reagent following the manufacturer's instructions (Dharmacon) and analyzed 48–96 h post transfection (Supplementary Table 2). Drug treatments were performed as follows: proteasome inhibitor MG-132 (Sigma) at 10 µM for 2–6 h, camptothecin (CPT; Sigma) at 1 µM for 1 h, cycloheximide (CHX; Sigma) at 20 µg ml⁻¹ for the indicated times, doxycyline (Sigma) at 5 µg ml⁻¹ for 48 h, leptomycin B (Sigma) at 20 nM for 3 h. For double-thymidine-block, cells were treated with 2–5 mM thymidine (Sigma) for 18 h, followed by 8–9 h release and a second, 15 h thymidine treatment. For CRISPR/Cas9-mediated USP21 deletion, TRI-DRGFP U2OS cells were transfected with a combination of three guide RNAs as outlined in Supplementary Fig. 2A. 48 h post transfection, cells were selected with 2 µg ml⁻¹ puromycin for 24 h followed by 48 h recovery. Cells were plated in 96-well plate via dilution cloning and single colonies were screened by PCR for USP21 genomic deletions as described in Supplementary Fig. 2A.

**HR and NHEJ assays**. HR efficiency was measured in U2OS cells carrying a Dox-inducible, TetR-regulated I-SceI transgene together with the DRGFP reporter (TRI-DRGFP cells)[27]. I-SceI expression was induced by adding 5 µg ml⁻¹ Dox. Repair of I-SceI-induced DSBs by HR results in a productive GFP reporter gene. No GFP⁺ cells were detected in the absence of Dox. NHEJ efficiency was measured using the pEJ5 reporter construct stably integrated into U2OS cells[31]. Two days after siRNA transfection or shRNA infection, pEJ5 cells were transiently transfected with an I-SceI expression vector to induce DSBs, a separate transfection with pEGFP-C1 was performed to normalize for transfection efficiency. HR or NHEJ efficiencies were analyzed by flow cytometry 48–72 h post DSB induction as the fraction of GFP⁺ cells.

**Clonogenic survival and oncosphere formation**. For clonogenic survival assays, HuH1 or Hep3B cells were infected with the indicated shRNA-expressing lenti-virus, counted 5–7 days after puromycin selection and plated in triplicate for each individual shRNA. HCC cell lines stably expressing Flag-tagged BRCA2 or a vector control plasmid were used for BRCA2 overexpression analyses. After 14–20 days, colonies were stained with 0.1% crystal violet in 20% methanol for 30 min[27]. Anchorage-independent growth was assessed by measuring HCC oncosphere formation[52]. Briefly, 75,000 cells were seeded on AlgiMatrix 3D Culture System (6-well plate, Life Technologies) and incubated for ~3 weeks to allow for spheroid formation. For sphere collection, the matrix was dissolved using AlgiMatrix Dissolving Buffer, spheres were pelleted at 300 g, resuspended with complete culture medium and reseeded, fixed in 100% ice-cold methanol and stained with 0.1% Crystal Violet. Sphere numbers were counted in triplicate. Images were taken before staining.

**Tumor xenografts**. Five- to 6-week-old male athymic nude mice (Crl:NUFoxn1nu, Charles River Laboratories) were injected subcutaneously with 2.5×10⁵ sh-Luc or sh-USP21-expressing HuH1 cells resuspended in 100 µl media with 100 µl Matrigel (BD Biosciences). Tumor sites were monitored weekly and subcutaneous nodule diameters were measured using a caliper. Mice were euthanized when largest tumors exceeded 1 mm³. All animal experiments were performed in compliance with guidelines provided by the National Cancer Institute (NCI) and the NIH Animal Care and Use Committee (ACUC).

**Antibodies**. The following antibodies were used for Western blotting (WB) and/or immunofluorescence (IF). Anti-53BP1 (sc-22760, Santa Cruz, WB 1:500, IF 1:300), anti-γ-H2AX (JBW301, Millipore, WB 1:5000, IF 1:300), anti-γ-H2AX (ab11174, Abcam, IF 1:300), anti-Flag (clone M2, Sigma, WB 1:5000, IF 1:500), anti-β-tubulin (CST2146S, Cell Signaling, WB 1:5000), anti-GAPDH (sc-32233, Santa Cruz, WB 1:1000), anti-GFP (sc-9996, Santa Cruz, WB 1:1000, IF 1:100), anti-HA (sc-805, Santa Cruz, WB 1:1000), anti-Myc (CST2272, Cell Signaling, WB 1:1000, IF 1:100), anti-USP21 (HPA028397, Sigma, WB 1:500–1:1000), anti-BRCA2 (OP95, Calbiochem, WB 1:1000), anti-BRCA1 (sc-SC-6954, Santa Cruz, IF 1:100), anti-RAD51 (sc-8349, Santa Cruz, WB 1:1000, IF 1:50), anti-CtIP (sc-271339, Santa Cruz, IF 1:100), anti-RPA (NA19L, Millipore, 1:200), FK2 (BML-PW8810-0100, Enzo, IF 1:000), anti-PALB2 (gift from B. Xia, WB 1:1000), anti-LaminA/C (sc-6215, Santa Cruz, WB 1:1000). The following secondary antibodies were used for WB: goat anti-rabbit HRP IgG (sc-2030, Santa Cruz, 1:5000), goat anti-mouse HRP IgG (sc-2031, Santa Cruz, 1:5000), donkey anti-goat HRP IgG (sc-2056, Santa Cruz, 1:1000). Secondary antibodies for IF were used at 1:250 dilution: Alex Fluor 488 anti-mouse IgG (H + L), Alexa Fluor 488 anti-Rabbit IgG (H + L), Alex Fluor 568 anti-mouse IgG (H + L), Alex Fluor 568 anti-Rabbit IgG (H + L).

**Immunoprecipitation**. Following indicated treatments, HEK293T cells were scraped and collected in lysis buffer (50 mM Tris pH 7.5, 150 mM NaCl, 0.5% NP40, 5 mM EDTA, protease inhibitor complete (Roche) and phosphatase inhibitor (Roche)). After lysis at 4 °C for 30 min, lysates were sonicated briefly and incubated with the following conjugated beads at 4 °C for 16 h: anti-HA agarose (Sigma), anti-Flag-M2 magnetic beads (Sigma), tandem ubiquitin binding entity 1-conjugated agarose-beads (TUBE1, Lifesensors). The beads were collected and washed with lysis buffer five times and boiled in sample buffer (0.4% SDS, 2% glycerol, 12 mM Tris-HCl pH 6.8, 0.1% (w/v) bromophenol blue, 5% (w/v) β-mercaptoethanol) prior to western blot. For denaturing IP, cells were transfected with HA-Ubiquitin and collected 72 h post transfection, following 3 h of MG-132 treatment. Cells were lysed in denaturing lysis buffer (20 mM Tris pH 8.0, 1% SDS, phosphatase/protease inhibitor) and briefly vortexed. Samples were heated at 95 °C for 5 min followed by addition of IP buffer (10 mM Tris pH 8.0, 150 mM NaCl, 1% Trition 100, 1 mM EDTA, phosphatase/protease inhibitor). Samples were passed through 20G needles to reduce viscosity. Following centrifugation, supernatant was collected for IP overnight at 4 °C. Protein A/G beads were added and incubated at 4 °C for 1 h. Following incubation, A/G beads were washed 4 times with wash buffer (20 mM Tris pH 8.0, 150 mM NaCl) and collected for WB.

**In vitro deubiquitination**. HEK293T cells were cotransfected with expression vectors for HA-ubiquitin and Flag-B6, Flag-B7 or Flag-BRCA2. Cells were collected 72 h post transfection following 6 h MG-132 treatment and lysed according to WB protocol. Lysate was batch-incubated with anti-Flag-M2 magnetic beads (Sigma) in lysis buffer diluted to 300 mM NaCl at 4 °C for 16 h. Flag beads were washed and eluted with 100 μg ml$^{-1}$ 3× Flag peptide (Sigma). Eluate was concentrated by centrifugation (Amicon Ultra-4 10 K) and buffer-exchanged into In Vitro DUB reaction buffer (50 mM NaCl, 50 mM Tris pH 7.4, 5 mM DTT). Recombinant USP21 (Abnova) was diluted with 10 × DUB activation buffer (1.5 M NaCl, 250 mM Tris pH 7.4, 100 mM DTT) and incubated at 23 °C for 10 minutes. For deubiquitination assays, 30 μl reactions were assembled with 20 ng μl$^{-1}$ USP21 and Flag-tagged substrate in DUB reaction buffer. Reactions were incubated at 30 °C for the indicated time periods, stopped with 6 × SDS loading buffer and analyzed by SDS-PAGE WB.

**Western blotting**. Cells were harvested 72 h after transfection or 7–14 days after puromycin selection. Whole cell lysate was prepared as described[27], and proteins were resolved by SDS-PAGE and transfered to a PVDF membrane (Millipore). Immuno-reactive proteins were detected by horseradish peroxidase-conjugated antibodies to rabbit or mouse and the ECL detection system (Amersham, RPN2132). Western blot images were acquired using a ChemiDoc imaging system (BioRad), quantification was performed using Image Lab (BioRad). Uncropped Western blot images are shown in Supplementary Fig. 8. For chromatin fractionation, nuclei were isolated in buffer A (10 mM HEPES (pH 7.9), 10 mM KCl, 1.5 mM MgCl$_2$, 0.34 M sucrose, 10% glycerol, 0.1% Triton X-100) with protease inhibitors (Roche) for 5 min on ice, lysates were pelleted at 1500 × g. Cytoplasmic supernatant was purified by high speed centrifugation to remove cell debris and insoluble aggregates. Nuclei were washed with buffer A and lysed at 4 °C for 30 min in 50 mM Tris pH 7.5, 400 mM NaCl, 0.5% NP40, 5 mM EDTA with protease inhibitors (Roche). Lysates were sonicated briefly prior to Western blot analysis.

**Immunofluorescence**. After indicated treatments, cells were fixed with 4% PFA in PBS and permeabilized with ice-cold methanol. For RAD51 foci staining, cells were pre-extracted twice with CSK buffer (10 mM Pipes, pH 7.0, 100 mM NaCl, 300 mM sucrose, 3 mM MgCl$_2$) containing 0.7% Triton X-100[60]. Following pre-extraction, cells were washed with PBS, fixed with 2% PFA and permeabilized with 0.2% Triton X-100. Cell staining was performed with the indicated primary and secondary antibodies. Confocal z-stacks were taken using a Zeiss LSM780

microscope. RAD51 foci from DAPI-segmented nuclear areas were quantified using ImageJ software.

**Laser microirradiation**. Laser microirradiation was performed on double-thymidine blocked cells using a Zeiss LSM780 confocal microscope equipped with a 364 nm UVA laser (Coherent). Cells were fixed in 4% PFA 30 mins post-irradiation and immunostained with the indicated antibodies. For γ-H2AX measurements, mean γ-H2AX intensity within damaged regions was normalized to sh-Luc control. For other DDR factor enrichment, damaged regions were defined by γ-H2AX staining. DDR factor intensity within damaged area was normalized against surrounding undamaged nuclear area.

**High-throughput imaging**. Immunofluorescence stained 96-well plates were fixed, permeabilized and stained as described above. Images were taken on an Opera QEHS high-throughput confocal microscope with Opera 2.0.1 software (Perkin-Elmer). The performance setting was planar apochromatic 40 X, NA 0.9 water immersion lens (Olympus) with 1.3-Mp CCD cameras (pixel binning of 2). The imaging configuration was 320 nm pixels. DAPI, Alexa488 and Alexa568 channels were acquired in three individual exposures. More than 300 cells were imaged per experimental condition. Images were analyzed using automated image analysis software (Acapella 2.6, Perkin-Elmer). Briefly, for every experimental condition, the DAPI channel was used to identify nuclei based on edge segmentation, then foci signals were counted based on a threshold set up according to the background determined from undamaged control samples. Values are expressed as foci per nucleus.

**Flow cytometry**. Immunostaining and flow cytometry analysis were performed following standard procedures, cleaved Caspase3-FITC was from BD Biosciences. For cell cycle analysis, cells were labeled with BrdU and stained using the FITC-BrdU Flow Kit (BD Biosciences).

**RNA extraction and quantitative real-time PCR**. Total RNA was extracted using the RNeasy Mini Kit (QIAGEN). cDNA was synthesized from 0.2 to 1 mg of total RNA using the ThermoScript RT-PCR system (Invitrogen), and expression of the indicated genes was analyzed by quantitative RT-PCR using a LightCycler 480 II (Roche), see Supplementary Table 1 for primer sequences.

**Statistics and survival analyses**. Students two-tailed t-tests were performed when there are two symmetrical sample sets. Wilcoxon rank-sum tests were performed for laser imaging analyses with non-Gaussian distributions, one-way ANOVA with Tukey's post hoc testing when comparing multiple groups. Kaplan-Meier curves were calculated using the Cox's proportional hazards model with a median cutoff. p-values <0.05 were considered to be significant. Sample sizes were selected on the basis of previous experiments and are listed in the Figure legends or respective Methods sections. Independent experiments were performed as indicated to guarantee reproducibility of findings. No randomization or blinding was used, and no animals or data points were excluded from analysis.

**TCGA data analyses**. For DUB copy number analysis, log2 values were extracted from Oncomine (https://www.oncomine.com/resource/login.html), data for 80 DUBs were available from a total of 139 HCC tumors. For USP21 gene expression analysis, level 3 TCGA-LIHC RNASeqV2 RSEM from 373 HCC tumor samples were downloaded from the public TCGA portal (https://tcga-data.nci.nih.gov/tcga/) and RSEM values of expressed DUBs were log2 transformed. To compare USP21 expression across cancer types, USP21 mRNA levels were analyzed for 26 different cancer types from a total of 10,205 primary tumor samples. Relative fold change was calculated between primary tumors and non-tumor tissues. Box plots in Supplementary Fig. 4A. were generated using R studio (Version 0.99.896).

**Microarray and arrayCGH analysis**. The data for the LCI cohort (n = 488) and the Laboratory of Experimental Carcinogenesis cohort (LEC, n = 180) are available on Gene Expression Omnibus (GEO). For clinical data and processing of microarrays from the LCI or LEC cohorts see the following refs [61, 62]. For SCNA data analysis, the arrayCGH data of 64 HCC samples from the LCI cohort was used. To look for genome instability, 64 HCC samples with SCNA data and mRNA expression data were divided into two groups based on USP21 mRNA levels using the median as a cutoff. Only segmented regions found in at least 20% of the 64 samples were considered to be significant aberrant changes. Regions with segmented log2 ratios greater than 0.5 and less than −0.5 were considered gains and losses, respectively.

**Data availability**. LCI and LEC cohort microarray and/or SCNA data were published previously and are available on GEO (LCI: GSE14322, GSE14520, LEC: GSE1898, GSE4024). Additional USP21 expression and copy number data were obtained from the public TCGA portal (https://tcga-data.nci.nih.gov/tcga/) and from Oncomine (https://www.oncomine.com/resource/login.html), respectively.

All other remaining data are available within the Article and Supplementary Files, or available from the authors upon request.

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

## Acknowledgements

We would like to thank S. Sharan for critical reading of the manuscript, T. Karpova and M. Kruhlak for imaging support, M. Jasin, H. Lee, B. Li, T. Misteli, S. Oberdoerffer, S. Sharan, A. Venkitaraman and B. Xia for reagents. High-content imaging acquisition and analysis was performed at the High-Throughput Imaging Facility (HiTIF)/Center for Cancer Research/NCI. This work was supported by the Intramural Research Program of the National Institutes of Health (NIH), NCI, Center for Cancer Research.

## Author contributions

P.O. and J.L. conceived the project. J.L. performed the TCGA data analysis, high-throughput imaging, IF, IP, western blot, clonogenic survival assays, xenograft and NHEJ assays. A.K. conducted flow cytometry, HR and NHEJ assays, western blot, in vitro deubiquitination, CRISPR analyses and clonogenic survival assays. H.D. performed the USP21 mRNA vs. copy number correlation, microarray and HCC patient survival analyses. A.D.T. performed laser microirradiation, RNA extraction and RT-PCR. S.M.K. performed the arrayCGH analysis. P.O. wrote the manuscript with help from J.L., H.D. and X.W.W.

## Additional information

**Competing interests:** The authors declare no competing financial interests.

