## [Peer Review File · Nature Communications]

Reviewers' comments:

Reviewer #1 (Remarks to the Author):

This m/s describes an interaction between BRCA2 and the deubiquitinating enzyme USP21. The data presented indicates that USP21 binds to and deubiquitinates BRCA2 leading to the stabilization of BRCA2. Consistent with this, depletion of USP21 reduces homologous recombination (HR) in a specific reporter assay, reduces BRCA2 protein levels and diminishes Rad51 focus formation, while modestly increasing the signal of gamma-H2AX, suggesting that the cells depleted of USP21 encounter increased levels of genotoxic damage. USP21 specifically binds to the DNA binding domain of BRCA2 and USP21 expression increases the abundance of fragments of BRCA2 that bind USP21, consistent with increased half life of these fragments when exposed to USP21's deubiquitinating activity. A DUB-defective point mutant of USP21, C221A, serves as a useful control. USP21 is shown to be highly expressed in HCC and suppression of USP21 suppresses growth of HCC cell lines. Some data suggests that BRCA2 overexpression can rescue this growth defect.

The data presented is internally consistent and generally persuasive. It remains quite unclear why USP21 is highly expressed in HCC cells. The idea that BRCA2 is the critical target of USP21 in HCC relies on a single experiment, in which overexpression of BRCA2 overcomes the growth defect of HCC cell lines depleted of USP21. This set of experiments is incomplete, since it is not shown what level of BRCA2 overexpression is obtained. The authors reason that the high levels of BRCA2 in HCC assist replication. However, it is not clear that BRCA2 levels are particularly high in HCC. Even if BRCA2 levels are elevated, it is possible that high USP21 expression in HCC drives cancer growth via its action on a different target. It would be helpful to address the interaction between USP21 and BRCA2 further by use of USP21 null cells (e.g., PMID 25680095). How do the authors explain the mild phenotype of USP21 null mice?

Specific points:

Did the authors normalize for transfection efficiency in the repair assays? This reviewer was unable to find this information in the methods section.

Fig 1: Define IQR (presumably "interquartile range"). What statistical test was used for box and whisker plots where the distribution was asymmetrical? In this case, a t-test may not be appropriate if the distribution does not fit a normal distribution.

USP21 depletion appears to reduce the S phase fraction in U2OS and HCC cultures. This effect could further reduce the level of HR detected in USP21-depleted cells.

Fig 2C control is not ideal. An HA-tagged USP21 mutant protein would be ideal. Does U21 C221A bind BRCA2? Can the authors define the binding site on USP21 for BRCA2?

Fig 3B The interpretation of the data would be strengthened if wtUSP21 overexpression were to increase the half life of BRCA2 fragments B7 and B8 (but not, for example, B6).

Fig 3C: Was this performed in the presence of proteasome inhibitors? If not, why are Ub species detectable?

Figs 4 and 5. As noted above, high expression of USP21 in HCC could be driven by its interaction with targets other than BRCA2. The authors' conclusion—that elevated BRCA2 alone is the critical driver of USP21 overexpression in HCC—is not the only way to interpret these results.

Fig 5G – the BRCA2 protein levels in the overexpression clones should be shown.

Reviewer #2 (Remarks to the Author):

In this manuscript by Liu et al. the authors identify the ubiquitin-specific protease USP21 as a positive regulator of BRCA2 stability. Author show USP21 interacts with BRCA2 resulting in decreased BRCA2 ubiquitination and stabilizes BRCA2 to promote efficient Rad51 loading at DSBs. Author also show that USP21 is overexpressed in hepatocellular carcinoma (HCC) resulting in increased BRCA2 stability and shorter survival.

BRCA1 and BRCA2 both play important roles in homologous recombination mediated DNA Double strand break (DSB) repair. Although the mutations in BRCA1 and BRCA2 are known to be critical for causing cancer very little is known about the over-expression phenotype. However, BRCA2 but not BRCA1 is found to be frequently overexpressed in sporadic cancers. For some unknown mechanisms, elevated BRCA2 expression favors cancer development while over-expression of BRCA1 does not. It is important to find out how and why BRCA2 is overexpressed. Therefore, this study describing BRCA2 stabilization by USP21 overexpression is interesting and potentially important.

The manuscript is written carefully. Most of the experiments are relatively well designed with proper controls. Data is largely of good quality and convincing. However, quality of western blot and constant loading control issues impact the inferences. Before concluding USP21 stabilizes BRCA2 to promote efficient Rad51 loading at DSBs by de-ubiquitinating BRCA2, the following concerns need to be addressed.

The major concern:

1. In supplement figure S2, shUSP21 results in increased 53BP1 and BRCA1 in DNA damage foci. If USP21 indeed stabilizes BRCA1, and USP21 is recruited to DNA damage foci, changes in BRCA2 damage foci after USP21 knock down or overexpression should be presented in form of Immunofluorescence staining and chromatin fractionation before and after DNA damage.
2. In figure 2, USP21 knock down results in decreased Rad51 and PalB2 protein level. Is Rad51 and PalB2 also substrate of USP21?
3. Poor quality of Western blot in supplement figure S3 makes the half-life quantification less convincing. Change in BRCA2 and Rad51 half-life is essential for the figure and therefore need to be presented in high quality in main figures instead of supplementary.
4. In supplement figure S2F, live cell imaging suggests USP21 recruited to damage foci peak in 4 minutes, but decreased almost completely after 10 minutes, however, increased ubiquitin at damage foci in USP21 knock down cells last for hours as shown in supplement S2A. Also half-life assay in supplemental figure S3A suggests only 50% of BRCA2 is degraded after 2 hours in shUSP21 cells, predicted effect of BRCA2 degradation in 10 minutes in damage foci is questionable. Half-life assay of chromatin associated BRCA2 after damage is needed.
5. Evidence for USP21 binding to BRCA2 and Rad51 is convincing. USP21 inhibits BRCA2 polyubiquitination is also supported by in vivo ubiquitination assay in figure S3. However, in order to demonstrate USP21 directly de-ubiquitinates BRCA2 but not Rad51 and PalB2, in vitro deubiquitination assay is needed. This would demonstrate USP21 alone is sufficient to remove ubiquitin from poly-ubiquitinated BRCA2 but not Rad51 and PalB2.
6. If shUSP21 indeed results in decreased BRCA2 in damage foci, delayed decrease of RPA foci and delayed increase of Rad51 foci should be observed in format of Figure S2ABC.
7. All experiments based on shRNA with can be due to off-target effect on Rad51 and gamma-H2AX and cell cycle distribution. Please reproduce some key data in USP21 CRISPR knock out cells or rescue USP21 knock down phenotype by overexpressing shRNA resistant USP21.

Minor concerns:

1. Title in Figure 1 legends states USP21 promotes HR by recruiting Rad51 to DSBs, this statement

is inconsistent with the discovery and may mislead readers.

2. Figure 1B, need western blot to show siRNA used in experiment indeed leads to decrease of USP21 and 53BP1 protein.
3. Figure 1CDEF, need FACS to show cell is indeed enriched S phase.
4. Figure 2B, FACS to show cell cycle or western blot show cell cycle markers.
5. Label in figure 2D input for poly-Ub is confusing.
6. Figure 3C, include result from USP21 knock down.
7. Supplemental figure S1D, cell cycle distribution seems to suggest cell cycle arrest in G1 in sh-U21-2 cell, please also show G1/G2.

Reviewer #3 (Remarks to the Author):

In this study, the authors investigated the role of USP21 in the DNA repair process and suggested that BRCA2 is a substrate of USP21. They showed that knockdown of USP21 resulted in decreased HR efficiency due to a reduced BRCA2 protein abundance. They found that USP21 is one of the most frequently amplified DUB in HCC, and oncogenic activities of USP21 contributed to HCC tumor growth. The authors propose the interesting model that USP21 is a component of signaling pathway associated with DNA repair, and BRCA2 is a positive regulator of liver cancers, however, the presented data are not compelling enough to fully support their claim.

Specific comments:

1. Figure 1B: Two independent siRNAs or shRNAs against USP21 are required.
2. Figure 2B: There is no correlation between BRCA2 and RAD51 abundance, which is not consistent with Figure 2A and Figure S3A-S3C.
3. Figure 2D: The BRCA2 and RAD51 WB panels need molecular weight markers.
4. Figure S3A: The panel and graphs should be moved to main figure, as this experiment is critical in proving the USP21-mediated posttranslational regulation of BRCA2. The graphs showing the quantification of the western blot analyses should have error bars (n = 3). In addition, a shorter time point (between 0 and 4 h upon CHX) should be included for an accurate half-life measurement.
5. Figure 3A-3B: Experiments in Figure 3A and 3B need to be repeated with a comparable expression level of each Flag-BRCA fragment.
6. Figure 3C: It would be better to include Flag-B6, B8, and B9 along with B7 for the assay to confirm that USP21 targets B7 and B8, but not B6 and B9, for deubiquitylation.
7. Figure 5G: The data is not consistent with Figure 5C. As a reference, sh-Luc cells transfected with empty vector needs to be included in the graph.
8. Figure 5C-5D and 5G: It is not enough to present the clonogenic survival assay data to claim the USP21 and BRCA2 oncogenic function in HCC. The authors need to evaluate anchorage independent growth in soft agar and tumorigenesis in nude mice.

Response to referees:

We would like to thank all reviewers for their valuable and constructive comments. We have now performed substantial experimental as well as textual revisions to address both the technical and conceptual concerns raised by the reviewers. The revised manuscript includes the following key improvements:

- 1.) Additional support for HCC relevance: To further corroborate the link between USP21 and BRCA2 in HCC tumor growth, we now show that BRCA2 loss causes a growth defect similar to that of USP21 knockdown (revised Fig. 5B). Moreover, we have extended our BRCA2-rescue studies as requested by reviewer 1 (revised Fig. 5E). Finally, we have added anchorage-independent growth and xenograft assays to demonstrate a role for USP21 function during tumor growth in more physiological conditions, as suggested by reviewer 3 (revised Fig. 5D, G).*
- 2.) Additional evidence that USP21 deubiquitinates BRCA2: We have now performed in vitro deubiquitination of both full-length BRCA2 and USP21-interacting BRCA2 fragments using recombinant USP21 enzyme and find that USP21 is sufficient to deubiquitinate these proteins in vitro (new Fig. S5). We have further extended our BRCA2-fragment analyses (revised Fig. 3C) and have substantiated BRCA2 protein stability analyses as requested by reviewers 2 and 3 (revised Fig. 2B).*
- 3.) Independent validation of USP21 knockdown: To exclude off-target effects of siRNA/shRNA analyses, we have generated USP21 KO cells via CRISPR/Cas9-based gene editing. Using these cells, we have confirmed the effect of USP21 loss on HR, BRCA2 stability and RAD51 recruitment.*

We believe that these modifications have significantly advanced the biological insight and conceptual impact of our work. Please find below a comprehensive, point-by-point response to the reviewers' comments.

Response to Reviewer #1:

This m/s describes an interaction between BRCA2 and the deubiquitinating enzyme USP21. The data presented indicates that USP21 binds to and deubiquitinates BRCA2 leading to the stabilization of BRCA2. Consistent with this, depletion of USP21 reduces homologous recombination (HR) in a specific reporter assay, reduces BRCA2 protein levels and diminishes Rad51 focus formation, while modestly increasing the signal of gamma-H2AX, suggesting that the cells depleted of USP21 encounter increased levels of genotoxic damage. USP21 specifically binds to the DNA binding domain of BRCA2 and USP21 expression increases the abundance of fragments of BRCA2 that bind USP21, consistent with increased half life of these fragments when exposed to USP21's deubiquitinating activity. A DUB-defective point mutant of USP21, C221A, serves as a useful control. USP21 is shown to be highly expressed in HCC and suppression of USP21

suppresses growth of HCC cell lines. Some data suggests that BRCA2 overexpression can rescue this growth defect.

The data presented is internally consistent and generally persuasive. It remains quite unclear why USP21 is highly expressed in HCC cells. The idea that BRCA2 is the critical target of USP21 in HCC relies on a single experiment, in which overexpression of BRCA2 overcomes the growth defect of HCC cell lines depleted of USP21. This set of experiments is incomplete, since it is not shown what level of BRCA2 overexpression is obtained. The authors reason that the high levels of BRCA2 in HCC assist replication. However, it is not clear that BRCA2 levels are particularly high in HCC. Even if BRCA2 levels are elevated, it is possible that high USP21 expression in HCC drives cancer growth via its action on a different target. It would be helpful to address the interaction between USP21 and BRCA2 further by use of USP21 null cells (e.g., PMID 25680095 MOUSE). How do the authors explain the mild phenotype of USP21 null mice?

Response: We would like to thank this reviewer for the positive evaluation of our manuscript and the helpful comments. We believe that the resulting revisions and additional experiments have significantly strengthened our conclusions.

The reviewer raises an important point regarding the use of USP21 null cells to address potential off target effects. We have, thus, inactivated USP21 via CRISPR/Cas9 in our U2OS HR-reporter cell line. USP21 deletion was confirmed at the genomic and protein levels (See Fig. S2 for KO strategy and validation). Using CRISPR KO clones, we were able to validate the impact of USP21 loss on HR, BRCA2 stability and RAD51 recruitment (see revised Figure 1B, H, and Fig. S2). Of note, USP21-deficient clones recovered from the HR defect after extended time in culture despite stable USP21 deletion (validated by genomic PCR and WB, see Fig. S2C, D). These findings point to compensatory mechanisms involving other DUBs and/or E3 ligases that may counteract deleterious consequences of USP21 loss. A similar observation has been reported in USP21-deficient mouse models (Jin et al., Nat Commun 2016; Pannu et al., PLoS ONE 2015) and is consistent with recent findings demonstrating genetic compensation induced by deleterious mutations but not gene knockdown (Rossi et al., Nature 2015, see also revised page 6). As a result, the majority of experiments were performed under conditions of acute USP21 depletion.

Regarding the nature of elevated USP21 expression in HCC, TCGA analyses showed that the USP21 gene is amplified in HCC and that copy number changes are in direct correlation with mRNA levels (see Fig. 4C), suggesting selection of an early gene amplification event. We have now discussed this further in the revised manuscript (page 12). We agree with the reviewer that increased USP21 expression in HCC is likely to affect the stability of proteins other than BRCA2, and we have emphasized this notion in the revised discussion (page 16). However, a role for BRCA2 in the observed tumor growth defect upon USP21 loss is supported both by the partial rescue upon BRCA2 overexpression (a Western blot is now included, see revised Fig. 5E), and by our analysis of CAPAN-1 cells, which carry a BRCA2 protein lacking the C-terminal, USP21-interacting domain and show little to no growth defect upon USP21 depletion (Fig. 5F). To further

corroborate a role for BRCA2 during HCC tumor growth, we have now determined the consequences of BRCA2 loss on clonogenic HCC survival and observe a growth defect comparable to USP21 knockdown (revised Fig. 5B). Together, these findings strongly support the notion that the survival defect upon USP21 loss in HCC cells can at least in part be attributed to the concomitant BRCA2 depletion.

Specific points:

Did the authors normalize for transfection efficiency in the repair assays? This reviewer was unable to find this information in the methods section.

Response: We apologize for the oversight. The HR assay utilized a cell line that carries a stably integrated, doxycycline-inducible I-SceI transgene, eliminating the need for endonuclease transfection (see Khurana et al., Cell Reports 2014). The transfection efficiency for the NHEJ assay was calculated using an independent transfection with peGFP. We have now clarified this in the methods section.

Fig 1: Define IQR (presumably “interquartile range”). What statistical test was used for box and whisker plots where the distribution was asymmetrical? In this case, a t-test may not be appropriate if the distribution does not fit a normal distribution.

Response: Following the reviewer's comment, P values have been recalculated using a two-sided Wilcoxon rank-sum test, which does not require a normal distribution. IQR has been defined as interquartile range.

USP21 depletion appears to reduce the S phase fraction in U2OS and HCC cultures. This effect could further reduce the level of HR detected in USP21-depleted cells.

Response: We agree that the observed reduction in S phase cells may contribute to the HR defect. However, given that the HR defect is significantly more pronounced than the loss of S phase cells, and in light of our analyses of repair defects in S phase cells (see Fig. 1D-H), we conclude that USP21 has a bona fide impact on HR beyond cell cycle control. Moreover, we have now generated CRISPR KO cells, in which we see a similar reduction in HR with minimal changes in cell cycle (see revised Fig. 1B, Fig. S2E).

Fig 2C control is not ideal. An HA-tagged USP21 mutant protein would be ideal. Does U21 C221A bind BRCA2? Can the authors define the binding site on USP21 for BRCA2?

Response: We agree with the reviewer that the identification of BRCA2 binding sites on USP21 would be useful, as this may allow us to separate between binding and catalytic activities of USP21. However, our preliminary analyses showed that USP21 interacts with BRCA2 fragments predominantly via its catalytic domain, precluding such separation of function analyses. We have thus not further investigated these fragments. With regard to Fig. 2C, we believe that the use of the WT USP21 protein represents a more

biologically relevant approach to investigate its potential to interact with BRCA2.

Fig 3B The interpretation of the data would be strengthened if wtUSP21 overexpression were to increase the half life of BRCA2 fragments B7 and B8 (but not, for example, B6).

Response: We thank the reviewer for this comment. However, we were unable to detect a reduction in the protein levels of overexpressed BRCA2 fragments even after extended CHX treatment (see Figure below). Given that prolonged CHX administration is cytotoxic, we were thus unable to assess USP21-mediated increase in fragment half lives.

Fig 3C: Was this performed in the presence of proteasome inhibitors? If not, why are Ub species detectable?

Response: We apologize for this oversight. Analyses in Figure 3C were performed in the presence of MG132 and this is now stated in the Figure legend. Please note that Figure 3C was replaced with an analysis of all four BRCA2 fragments tested in Fig. 3B, demonstrating that B7 and B8 are the predominant, ubiquitinated species.

Figs 4 and 5. As noted above, high expression of USP21 in HCC could be driven by its interaction with targets other than BRCA2. The authors' conclusion—that elevated BRCA2 alone is the critical driver of USP21 overexpression in HCC—is not the only way to interpret these results.

Response: We agree with the reviewer that, while our findings identify BRCA2 as an important target for USP21 function in HCC, there are likely to be additional enzymatic targets. We apologize if it was not sufficiently emphasized and have now revised the discussion accordingly (see page 16).

Fig 5G – the BRCA2 protein levels in the overexpression clones should be shown.

Response: We have now independently validated the partial rescue of clonogenic survival by BRCA2 overexpression, and have included a Western blot analysis for BRCA2 levels in this experiment (revised Fig. 5E)

Response to Reviewer #2:

In this manuscript by Liu et al. the authors identify the ubiquitin-specific protease USP21 as a positive regulator of BRCA2 stability. Author show USP21 interacts with BRCA2 resulting in decreased BRCA2 ubiquitination and stabilizes BRCA2 to promote efficient Rad51 loading at DSBs. Author also show that USP21 is overexpressed in hepatocellular carcinoma (HCC) resulting in increased BRCA2 stability and shorter survival.

BRCA1 and BRCA2 both play important roles in homologous recombination mediated DNA Double strand break (DSB) repair. Although the mutations in BRCA1 and BRCA2 are known to be critical for causing cancer very little is known about the over-expression phenotype. However, BRCA2 but not BRCA1 is found to be frequently overexpressed in sporadic cancers. For some unknown mechanisms, elevated BRCA2 expression favors cancer development while over-expression of BRCA1 does not. It is important to find out how and why BRCA2 is overexpressed. Therefore, this study describing BRCA2 stabilization by USP21 overexpression is interesting and potentially important.

Response: We thank the reviewer for the positive and constructive comments. We have now addressed the concerns raised below and feel that these additions significantly strengthen our conclusions. Please see below for a detailed response.

The manuscript is written carefully. Most of the experiments are relatively well designed with proper controls. Data is largely of good quality and convincing. However, quality of western blot and constant loading control issues impact the inferences. Before concluding USP21 stabilizes BRCA2 to promote efficient Rad51 loading at DSBs by de-ubiquitinate BRCA2, the following concerns need to be addressed.

The major concern:

1. In supplement figure S2, shUSP21 result in increased 53BP1 and BRCA1 in DNA damage foci. If USP21 indeed stabilizes BRCA1, and USP21 is recruited to DNA damage foci, changes in BRCA2 damage foci after USP21 knock down or overexpression should be presented in form of Immuno-Florescence staining and chromatin fractionation before and after DNA damage.

Response: The reviewer raises an important point and we apologize for not sufficiently addressing this issue in the original submission. Our data indicate that USP21 affects overall BRCA2 stability rather than DSB site-specific BRCA2 accumulation. Consistent with this, we observe reduced steady state levels of BRCA2 in S phase cells even in the absence of DSB induction (Fig. 2C), which limits available BRCA2 in this HR-permissive cell cycle phase. Importantly, following this reviewer's suggestion, we now show that the latter translates into a reduction in chromatin-associated BRCA2 upon S phase damage, which is consistent with the HR defect observed in USP21 depleted cells (see revised Fig. S4A).

2. In figure 2, USP21 knock down result in decreased Rad51 and PalB2 protein level. Is Rad51 and PalB2 also substrate of USP21?

Response: We have now analyzed the impact of USP21 overexpression on PALB2 and RAD51 ubiquitination to address this comment. Using TUBE1-based immunoprecipitation, which has a 1000 X greater affinity for poly-ubiquitin compared to mono-ubiquitin, we observe no evidence for high molecular weight ubiquitin chains on either PALB2 or RAD51. TUBE1-based co-purification of PALB2 and RAD51 was thus likely the result of complex formation with poly-ubiquitinated BRCA2. Consistent with this, overexpression of WT USP21 but not the catalytic mutant resulted in a marked reduction of BRCA2-Ub as well as its interactors RAD51 and PALB2 (see revised Fig. 2E).

3. Poor quality of Western blot in supplement figure S3 makes the half-life quantification less convincing. Change in BRCA2 and Rad51 half-life is essential for the figure and therefore need to be presented in high quality in main figures instead of supplementary.

Response: The reviewer raises an important point and we have now repeated protein half-life analyses for BRCA2 and RAD51 using four independent infections with sh-Luc/sh-U21-1. We have further included a 2 h timepoint (see new Western blot in Fig. 2B). These analyses confirm the impact of USP21 loss on the stability of both proteins and are now shown in the main text (revised Fig. 2B).

4. In supplement figure S2F, live cell imaging suggest USP21 recruited to damage foci peak in 4 minutes, but decreased almost completely after 10 minutes, however, increased ubiquitin at damage foci in USP21 knock down cells last for hours as shown in supplement S2A. Also half-life assay in supplemental figure S3A suggest only 50% of BRCA2 is degraded after 2 hours in shUSP21 cells, predicted effect of BRCA2 degradation in 10 minutes in damage foci is questionable. Half-life assay of chromatin associated BRCA2 after damage is needed.

Response: We thank the reviewer for this comments and would like to clarify these issues. Due to the differences in damage load and DSB density, it is difficult to directly compare repair factor kinetics at sites of laser induced damage with foci formation at IRIFs, which are thought to present single DNA lesions (see review by Polo and Jackson, PMID 21363960). Nevertheless, we see a dynamic and transient increase in FK2 signal at foci that qualitatively matches the dynamic recruitment at laser stripes. Moreover, transient action of DUBs may have lasting effects on downstream repair factors, e.g by counteracting E3 ligases, some of which are known to be transiently and dynamically recruited to DSBs (e.g. Yan et al., MCB 2013, PMID: 23230272). However, given that our data point to a role for USP21 in HR downstream of BRCA1/53BP1 recruitment as well as end resection (see intact RPA recruitment in Fig. 1F), an in depth analysis of a possible

role for USP21 during BRCA1/53BP1 recruitment at DSBs is beyond the scope of this manuscript.

Regarding the kinetics of BRCA2 stabilization, our data indicate that this effect is not restricted to the presence of DNA damage (see response to point 1). We, therefore, believe that the extended half life analyses for total BRCA2 in revised Figure 2B sufficiently address the impact of USP21 on BRCA2 stability.

5. Evidence for USP21 binding to BRCA2 and Rad51 is convincing. USP21 inhibit BRCA2 polyubiquitination is also supported by in vivo ubiquitination assay in figure S3. However, in order to demonstrate USP21 directly de-ubiquitinates BRCA2 but not Rad51 and PalB2, in vitro deubiquitination assay is needed. This would demonstrate USP21 alone is sufficient to remove ubiquitin from poly-ubiquitinated BRCA2 but not Rad51 and PalB2.

Response: We agree with this comment and have now performed in vitro deubiquitination assays using recombinant USP21 protein (Abnova, catalog # H00027005-P01). Flag-tagged substrates were expressed together with HA-Ub and purified by immunoprecipitation. We observed robust removal of HA-Ub from Flag-BRCA2 fragments, which is entirely dependent on the addition of recombinant USP21. Similar results were observed for full length Flag-BRCA2 (see new Fig. S5). These findings demonstrate that USP21 is sufficient to deubiquitinate BRCA2 fragments. Given that we were unable to observe robust polyubiquitination of PALB2 and RAD51 in our in vivo assay (see Fig. 2E), we did not further pursue these proteins in vitro.

6. IF shUSP21 indeed result in decreased BRCA2 in damage foci, delayed decrease of RPA foci and delayed increase of Rad51 foci should be observed in format of Figure S2ABC.

Response: We agree with this prediction. However, we believe that the data reported in Fig. 1E-H are sufficient to conclude a role for USP21 in RAD51 accumulation at DSBs downstream of RPA. Importantly, we have now further confirmed impaired RAD51 loading using CRISPR-based USP21 KO cells (revised Fig. 1H, see also response to point 7).

7. All experiment based on shRNA with can be due to off-target effect on Rad51 and gamma-H2AX and cell cycle distribution. Please reproduce some key data in USP21 CRISPR knock out cells or rescue USP21 knock down phenotype by overexpressing shRNA resistant USP21.

Response: The reviewer raises an important point. We have now reproduced key data using USP21 CRISPR KO cell lines to address the possibility of off target effects and corroborate our findings from knockdown cells. These data confirm the impact of USP21 on HR, BRCA2 protein levels and Rad51 loading (see revised Fig. 1B, H, new Fig. S2).

Minor concerns:

1. Title in Figure 1 legends states USP21 promotes HR by recruiting Rad51 to DSBs, this statement is inconsistent with the discovery and may mislead readers.

Response: We have changed this title to “ USP21 promotes HR by facilitating RAD51 accumulation at DSBs”.

2. Figure 1B, need western blot to show siRNA used in experiment indeed leads to decrease of USP21 and 53BP1 protein.

Response: We have included a Western blot to validate siRNA-mediated knockdown (revised Fig. S1D).

3. Figure 1CDEF, need FACS to show cell is indeed enriched S phase.

Response: We have added a FACS analysis of DNA content following double thymidine block, demonstrating that > 80 % of cells were in S/G2 in control and USP21 KD cells (see revised Fig. S1F).

4. Figure 2B, FACS to show cell cycle or western blot show cell cycle markers.

Response: See response to point 3.

5. Label in figure 2D input for poly-Ub is confusing.

Response: We have removed “Poly-Ub” from the figure to avoid confusion.

6. Figure 3C, include result from USP21 knock down.

Response: MG132 treatment allows for maximal accumulation of ubiquitinated proteins. We thus feel that the use of catalytic dead and WT USP21 and the concomitant assessment of a loss of poly-Ub provides more conclusive insight than the depletion of USP21.

7. Supplemental figure S1D, cell cycle distribution seems to suggest cell cycle arrest in G1 in sh-U21-2 cell, please also show G1/G2.

Response: We have now added this information. Please note that the change in the fraction of HR-relevant S/G2 cells is significantly less pronounced than the reduction in HR, corroborating a bona fide HR defect, as discussed in the manuscript. Moreover, USP21 CRISPR KO cells show a similar HR defect despite comparable S phase fractions (new Fig. S2E).

Response to Reviewer #3:

In this study, the authors investigated the role of USP21 in the DNA repair process and suggested that BRCA2 is a substrate of USP21. They showed that knockdown of USP21 resulted in decreased HR efficiency due to a reduced BRCA2 protein abundance. They found that USP21 is one of the most frequently amplified DUB in HCC, and oncogenic activities of USP21 contributed to HCC tumor growth. The authors propose the interesting model that USP21 is a component of signaling pathway associated with DNA repair, and BRCA2 is a positive regulator of liver cancers, however, the presented data are not compelling enough to fully support their claim.

Response: We thank the reviewer for the constructive comments. We have now addressed the concerns raised below and feel that these additions significantly strengthen our conclusions.

Specific comments:

1. Figure 1B: Two independent siRNAs or shRNAs against USP21 are required

Response: We have added the two USP21 shRNAs used throughout the manuscript to the NHEJ analysis (see revised Fig. S1E). These results corroborate our finding that USP21 depletion predominantly affects DSB repair by HR.

2. Figure 2B: There is no correlation between BRCA2 and RAD51 abundance, which is not consistent with Figure 2A and Figure S3A-S3C.

Response: We have now performed extended protein stability analyses (see point 4), further corroborating a correlation between USP21, BRCA2 and RAD51 protein levels (see revised Fig. 2B, see also Magwood et al., DNA Repair 2013). Variable effects of USP21 loss on RAD51 protein levels are likely to reflect cell cycle state and/or compensatory RAD51 expression, pointing to a more complex relationship between BRCA2 and RAD51 (see also discussion, page 15). We hope this reviewer agrees that a detailed dissection of the latter is beyond the scope of this manuscript.

3. Figure 2D: The BRCA2 and RAD51 WB panels need molecular weight markers.

Response: MW markers have been added.

4. Figure S3A: The panel and graphs should be moved to main figure, as this experiment is critical in proving the USP21-mediated posttranslational regulation of BRCA2. The graphs showing the quantification of the western blot analyses should have error bars (n = 3). In addition, a shorter time point (between 0 and 4 h upon CHX) should be included for an accurate half-life measurement.

Response: We have now included the analysis of replicates from four independent shRNA

infections for the 4 h and 8 h time points. Following the reviewer's suggestion, a 2h time point has also been included, further corroborating a role for USP21 in the posttranslational regulation of BRCA2 (see revised Fig. 2B).

5. Figure 3A-3B: Experiments in Figure 3A and 3B need to be repeated with a comparable expression level of each Flag-BRCA fragment.

Response: While we agree with the reviewer that equal protein levels would visually improve these panels, our revised panel 3C shows that fragments B6 and B7 are subject to significant polyubiquitination, which likely provides a biological basis for the difference in overall protein levels. Stabilization of B7 and B8, e.g. by inhibiting proteasomal degradation, would preclude measurements of protein stability changes in Fig. 3B. Importantly, however, HA-USP21, which was used as the primary target for immunoprecipitation, was expressed at comparable levels (see Fig. 3A).

6. Figure 3C: It would be better to include Flag-B6, B8, and B9 along with B7 for the assay to confirm that USP21 targets B7 and B8, but not B6 and B9, for deubiquitylation.

Response: We agree with the reviewer and have repeated this assay with all four BRCA2 fragments to confirm that ubiquitination and USP21-mediated deubiquitination is most pronounced for fragments 7 and 8 (see revised Fig. 3C).

7. Figure 5G: The data is not consistent with Figure 5C. As a reference, sh-Luc cells transfected with empty vector needs to be included in the graph.

Response: We apologize if the description of this panel was not clear. The bar graph shows colony formation frequency in USP21 knockdown samples relative to sh-Luc-infected cells, which was set as 100 % for both control or BRCA2-overexpressing cells. This is now clarified this in the figure legend.

8. Figure 5C-5D and 5G: It is not enough to present the clonogenic survival assay data to claim the USP21 and BRCA2 oncogenic function in HCC. The authors need to evaluate anchorage independent growth in soft agar and tumorigenesis in nude mice.

Response: We agree with the reviewer that these are important additional experiments. We have now assessed anchorage-independent growth in vitro as well as tumor formation upon xenograft in nude mice using HuH1 hepatoma cells. The results support our conclusion that USP21 can promote HCC tumor growth (revised Fig. 5D, G). To assess anchorage-independent growth, we performed 3D HCC organoid culture in the presence or absence of USP21, using an Aligimatrix-based approach, which we have recently established and found to effectively reflect HCC aggressiveness in vitro (Takai et al, Scientific Reports, 2016).

Reviewers' comments:

Reviewer #1 (Remarks to the Author):

The concerns raised previously have been addressed satisfactorily and I recommend acceptance of the m/s.

Reviewer #2 (Remarks to the Author):

The authors have addressed several concerns and included new high quality data to support their hypothesis that USP21 stabilize BRCA2 to promote efficient Rad51 loading at DSBs by deubiquitinate BRCA2. However, the weakness of the study is the inability to demonstrate endogenous USP21 is responsible for maintaining BRCA2 stability and exclude PalB2 as the real target. Overexpressed USP21 can deubiquitinate BRCA2 in vitro, but the half-life change of BRCA2 is not significant. There is also lack of evidence of change in BRCA2 on damaged foci, USP21 localization to damage foci for less than 10 minutes is inconsistent with the fact that USP21 binds and protects BRCA2 for hours after damage.

Two specific issues:

Figure 2E did not demonstrate evidence for high molecular weight ubiquitin chains on BRCA2 either, TUBE1-based co-purification of PALB2 and RAD51 was thus likely the result of complex formation with poly-ubiquitinated unidentified protein in cell lysate.

The dynamic and transient increase in FK2 signal at foci can be any ubiquitinated protein, including PalB2 and Rad51. BRCA1 and BRCA2 are both E3 ubiquitin ligase, FK2 signal can be any of their substrates. The bottom line of the paper point to decreased BRCA2 protein, but not PalB2 or Rad51 protein, at damage foci is due to lack of USP21 protein to binding and deubiquitination and stabilization of BRCA2. Regarding the kinetics of BRCA2 stabilization, most data provided indicates that this effect is not restricted to the presence of DNA damage, Figure S4A suggest only small decrease in chromatin bound BRCA2 in the absence of treatment.

Reviewer #3 (Remarks to the Author):

The authors have addressed the questions in a satisfactory manner.

Response to reviewers:

We would like to thank all reviewers for their positive evaluation of the revised manuscript. A detailed response to the remaining concerns follows below.

Reviewer #1 (Remarks to the Author):

The concerns raised previously have been addressed satisfactorily and I recommend acceptance of the m/s.

Response: *We are happy to hear that we have satisfactorily addressed all concerns and thank this reviewer for the recommendation to accept our manuscript.*

Reviewer #2 (Remarks to the Author):

The authors have addressed several concerns and included new high quality data to support their hypothesis that USP21 stabilize BRCA2 to promote efficient Rad51 loading at DSBs by deubiquitinate BRCA2. However, the weakness of the study is the inability to demonstrate endogenous USP21 is responsible for maintaining BRCA2 stability and exclude PalB2 as the real target. Overexpressed USP21 can deubiquitinate BRCA2 in vitro, but the half-life change of BRCA2 is not significant. There is also lack of evidence of change in BRCA2 on damaged foci, USP21 localization to damage foci for less than 10 minutes is inconsistent with the fact that USP21 binds and protects BRCA2 for hours after damage.

Response: *We would like to thank this reviewer for the positive evaluation of our new data and have revised the manuscript to address the remaining concerns. To corroborate that BRCA2 loss occurs at the level of protein stability, we have performed additional, independent CHX analyses, which further underline the significance of the decrease in BRCA2 protein stability upon USP21 depletion (see revised Fig. 2B). Moreover, we were able to conclusively show that endogenous BRCA2 is poly-ubiquitinated in vivo using a denaturing IP approach (see revised Fig. S5A and our response to comment 1 below), providing an important addition to our in vitro studies (Figs. S5B-D). Detection of altered BRCA2 ubiquitination upon depletion of endogenous USP21 is complicated by the fact that poly-ubiquitinated BRCA2 is subject to degradation, and, while inhibition of proteasomal degradation by MG-132 allows for maximal accumulation of ubiquitinated proteins, it is likely to obscure any dynamic difference in ubiquitination between USP21 knockdown and control samples.*

Together with our extensive previous analyses, involving USP21 CRISPR knockout and knockdown approaches in a variety of different tumor cell lines, as well as the biochemical dissection of USP21-mediated BRCA2 deubiquitination (Fig. 3), we believe that these revisions provide compelling evidence that BRCA2 levels are controlled by ubiquitination and that loss of endogenous USP21 protein promotes decreased BRCA2 protein stability. The identification of BRCA2 as a central target for USP21-modulated protein stability is in agreement with a key paper demonstrating control of BRCA2 function by its interactor PALB2 (Ref. 24, Xia et al., Mol

Cell 2006). Among other important observations, this report demonstrates that BRCA2 is significantly less stable than PALB2 and that BRCA2 but not PALB2 can be readily targeted for proteasomal degradation. Indeed, CHX analyses show no loss in PALB2 levels over a period of 6 h (Xia et al, Fig. 3 G), consistent with the lack of poly-ubiquitinated PALB2 species in our TUBE IP (Fig. 2E). This work is now discussed on pages 9 and 15. We, nevertheless, acknowledge the possibility of additional BRCA2 and/or HR-related USP21 targets (see page 15).

Regarding USP21 function at sites of damage, the latter is likely to reflect a distinct role for USP21, which causes transient modulation of break-proximal ubiquitin levels and a concomitant increase in the recruitment of ubiquitin-dependent repair factors including BRCA1 (Jackson and Durocher, Mol Cell 2013). As we discuss in the manuscript, this effect is unlikely to explain the HR defect, as recruitment of BRCA1 and CtIP as well as end resection are not impaired in USP21-depleted cells. In contrast to DSB-proximal effects of USP21, its impact on BRCA2 stability is not limited to sites of damage and can be observed even in the absence of DSB induction, as is apparent from our protein stability analyses and other assays in Figure 2. As a result, cells are unable to efficiently perform BRCA2-dependent repair (including RAD51 loading at DSBs) when they encounter damage. The latter is supported by the observed decrease in chromatin-associated BRCA2, which was added in response to the reviewer's previous helpful comment.

Two specific issues:

Figure 2E did not demonstrate evidence for high molecular weight ubiquitin chains on BRCA2 either, TUBE1-based co-purification of PALB2 and RAD51 was thus likely the result of complex formation with poly-ubiquitinated unidentified protein in cell lysate.

Response: We agree that, due to the high molecular weight of BRCA2, poly-ubiquitinated BRCA2 species are difficult to distinguish from unmodified BRCA2. To conclusively demonstrate that BRCA2 is directly ubiquitinated in vivo, we have, therefore, performed IP for endogenous BRCA2 in the presence of HA-Ub, followed by HA Western blot. To eliminate interference of BRCA2-interacting proteins, this IP assay was done under denaturing conditions. Revised Fig. S5A shows that endogenous BRCA2 is efficiently poly-ubiquitinated in MG-132 treated cells in vivo.

The dynamic and transient increase in FK2 signal at foci can be any ubiquitinated protein, including PalB2 and Rad51. BRCA1 and BRCA2 are both E3 ubiquitin ligase, FK2 signal can be any of their substrates. The bottom line of the paper point to decreased BRCA2 protein, but not PalB2 or Rad51 protein, at damage foci is due to lack of USP21 protein to binding and deubiquitination and stabilization of BRCA2. Regarding the kinetics of BRCA2 stabilization, most data provided indicates that this effect is not restricted to the presence of DNA damage, Figure S4A suggest only small decrease in chromatin bound BRCA2 in the absence of treatment.

Response: The reviewer is absolutely correct in that the effect of USP21 loss on BRCA2 does not appear to be restricted to the presence of damage (see above). The effect on FK2 signal at DSBs, on the other hand, is likely a complementary and/or independent, break-associated function of

USP21. This notion is supported by the fact that USP21 recruitment and concomitant modulation of DSB-proximal ubiquitin are transient early events, as well as by the finding that we readily detect BRCA1 foci formation and end resection, which together fails to explain the observed HR defect (see also page 7) and led us to investigate downstream events.

Reviewer #3 (Remarks to the Author):

The authors have addressed the questions in a satisfactory manner.

Response: We thank the reviewer once again for the helpful comments, which have significantly improved the manuscript.

REVIEWERS' COMMENTS:

Reviewer #2 (Remarks to the Author):

The authors have addressed my concerns.